# AN ASYNCHRONOUS BUNDLE METHOD FOR DISTRIBUTED LEARNING PROBLEMS

**Daniel Cederberg**
Stanford University, USA

**Xuyang Wu**
SUSTech, China

**Stephen Boyd**
Stanford University, USA

**Mikael Johansson**
KTH, Sweden

## ABSTRACT

We propose a novel asynchronous bundle method for solving distributed learning problems. Compared to several existing asynchronous optimization algorithms, our method computes the next iterate based on a more accurate approximation of the objective function, and does not require any prior information about the maximal information delay in the system. This makes the proposed method fast and easy to tune. We prove that the algorithm converges in both deterministic and stochastic (mini-batch) settings, and quantify how the convergence rates depend on the level of asynchrony. The practical advantages of our method are illustrated through numerical experiments on classification problems of varying complexities and scales.

## 1 INTRODUCTION

We consider a setting where data is distributed among $n$ workers, each with its own smooth convex loss function $f_i : \mathbf{R}^d \to \mathbf{R}$. Our goal is to compute a solution $x^\star$ of

$$\text{minimize } F(x) \triangleq f(x) + R(x), \tag{1}$$

where $f(x) \triangleq \sum_{i=1}^n f_i(x)$ and $R : \mathbf{R}^d \to \mathbf{R}$ is a (possibly non-smooth) proper closed and convex regularizer. This problem template is ubiquitous in machine learning and includes lasso (Tibshirani, 1996), logistic regression (Koh et al., 2007), and many other important problem classes.

When data is distributed among multiple workers, algorithms that require synchronization at every iteration are limited by the slowest worker, creating a bottleneck. *Asynchronous* algorithms (Bertsekas & Tsitsiklis, 1989; Assran et al., 2020) address this issue by relaxing synchronization constraints, potentially resulting in machine learning systems that are both faster and easier to implement than their synchronous counterparts (Hannah & Yin, 2017). However, designing asynchronous algorithms is challenging because information computed at the workers — such as function values and gradients — may be obsolete by the time it reaches some coordinating mechanism like a central server. As a result, convergence guarantees for asynchronous algorithms often rely on an upper bound on the information delay from the workers, which is typically large and difficult to determine. Moreover, the step sizes permitted by these guarantees tend to shrink quickly as the delay bound increases. This complicates the implementation of asynchronous algorithms: if the estimated delay bound is too small, it may not hold, invalidating the theoretical convergence guarantees. Conversely, setting the bound too large results in overly conservative step sizes and slow practical convergence.

The design of most optimization algorithms relies on a *simple* approximation of the objective function — often referred to as a *model* in the optimization literature. For example, gradient descent is based on the quadratic upper bound of an $L$-smooth convex function, while the Polyak step size (Polyak, 1964) relies on a piecewise linear lower bound for a convex function with a known optimal value. However, a string of recent papers on *synchronous* algorithms (Davis & Drusvyatskiy, 2019; Asi & Duchi, 2019; Nesterov & Florea, 2021) suggests that using a more accurate approximation of the objective to compute the next iterate can improve performance. This naturally leads to the question of whether *asynchronous* algorithms could also benefit from more accurate approximations of the objective, potentially resulting in asynchronous algorithms with faster practical convergence and simpler tuning.

**Contributions**. We propose a parallel and asynchronous optimization method that leverages a more accurate approximation of the objective function to compute the next iterate. The method is de-

signed for a parameter server architecture (Li et al., 2013) and decouples gradient evaluations at the workers from decision vector updates at the master, making it robust to system asynchrony. On the theoretical side, we prove that our algorithm converges for all bounded delays and can be implemented without knowledge of the maximum delay. (We only know of two other asynchronous algorithms (Mishchenko et al., 2018; Wu et al., 2022) that are designed for a parameter server and share these properties while also explicitly taking the regularizer $R(x)$ into account.) On the practical side, we present an algorithm that converges quickly with minimal tuning. Our method supports stochastic function and gradient evaluations and can be viewed as an asynchronous bundle method, generalizing algorithms in Nesterov & Florea (2021) and Asi & Duchi (2019) to an asynchronous setting.

**Outline.** The paper is structured as follows. In §2, we relate our contribution to existing work. In §3, we introduce our asynchronous model-based algorithm, followed by a convergence analysis in §4. Implementation details and numerical experiments are presented in §5 and §6, respectively. Finally, we summarize our findings and conclude in §7.

## 2 RELATED WORK

**Model-based optimization.** Model-based optimization is a general framework in which an approximation, or *model*, of the objective function is maintained and used to compute the next iterate. This framework encompasses several well-known methods and principles, including the expectation-maximization algorithm (Dempster et al., 1977; Neal & Hinton, 1998), quasi-Newton methods (Dennis & Moré, 1977), bundle methods (Mäkelä, 2002), the majorization-minimization principle (Mairal, 2015; Lange, 2016), and acceleration techniques (d'Aspremont et al., 2021). In stochastic optimization, recent research has shown that using more accurate approximations of the objective function can improve both speed and robustness to step size selection (Duchi & Ruan, 2018; Davis & Drusvyatskiy, 2019; Asi & Duchi, 2019). For composite non-stochastic optimization, Nesterov & Florea (2021) recently demonstrated that constructing a piecewise linear model of the smooth part of the objective — rather than relying solely on the most recent gradient to approximate the smooth part — can lead to better performance. Our work aims to extend the idea of more accurate objective function models to asynchronous optimization.

**Parallel and asynchronous optimization.** For parallel optimization with a parameter server architecture, asynchronous algorithms can significantly outperform their synchronous counterparts (Hannah & Yin, 2017). They have also shown promising results for other architectures (Recht et al., 2011; Chaturapruek et al., 2015). Many asynchronous methods use diminishing step sizes (Duchi et al., 2015; Assran & Rabbat, 2020) or rely on a predetermined maximum iteration number (Koloskova et al., 2022; Mishchenko et al., 2022; Recht et al., 2011). Exceptions that are well-suited for a parameter server generally fall into two categories: (1) methods that require knowledge of an upper bound on information delays (Zhang & Kwok, 2014; Peng et al., 2016; Gürbüzbalaban et al., 2017; Vanli et al., 2018; Wai et al., 2020; Sun et al., 2019), and (2) algorithms that do not rely on such a bound (Feyzmahdavian et al., 2014; Mishchenko et al., 2018; 2020; Wu et al., 2022; 2023). In practice, however, an upper bound on the information delay is often unknown in advance. Since the admissible step sizes shrink as the upper delay bound increases, it is often difficult to guarantee that the algorithms in the first group converge in practice.

Most of the asynchronous methods above perform simple, closed-form updates at the central server, while the bulk of the computational work (such as gradient and function evaluations) is offloaded to the workers. The reason for the cheap update at the central server is, using the terminology of model-based optimization, that the server maintains a *simple* model of the objective function based only on the *most recent* information from each worker. Our approach differs in that we propose using a more accurate model of the objective function — one that incorporates more than just the latest information from each worker. The fact that computationally intensive tasks are handled by the workers suggests an opportunity to investigate algorithms that are slightly more complex at the server, like the one we propose, to potentially improve the overall system performance.

**Bundle methods.** A key challenge in asynchronous optimization is that gradients provide *local descent directions*, making it challenging to combine gradients from different workers computed at different iterates into a meaningful search direction that ensures descent. In contrast, gradients (together with function values and convexity) provide *global lower bounds* on the objective func-

tion, making it easier to combine gradients evaluated at widely different points into a valid lower bound of the objective function. This observation motivates our method, which can be viewed as an *asynchronous bundle method*.

In the *non-smooth* optimization literature, several versions of asynchronous bundle methods have been proposed (Emiel & Sagastizábal, 2010; Iutzeler et al., 2020; van Ackooij & Frangioni, 2018; de Oliveira & Eckstein, 2015). Since non-smooth optimization covers a wide range of problems, the convergence results for existing asynchronous bundle methods are generally weak. (For example, a typical convergence result is that any cluster point of the iteration sequence solves the problem (Emiel & Sagastizábal, 2010, Proposition 4).) However, our setting in this paper is different. While the aforementioned works assume that the non-smoothness of the objective is present in the finite-sum structure, we assume that the finite-sum structure arises in a smooth part of the objective, and the non-smoothness of the objective is caused by a regularizer. This key distinction allows us to derive much stronger convergence guarantees — not only do we prove convergence, but we also derive convergence rates (see Theorem 4.5 and Theorem 4.9).

Bundle methods have also received direct attention from the machine learning community (Teo et al., 2007; Franc & Sönnenburg, 2009; Teo et al., 2010; Chu et al., 2017; Paren et al., 2022). The first three works design variants of bundle methods for general empirical risk minimization, but they differ from our method in several ways. (For example, these methods maintain a piecewise linear model of the sum $f = \sum_{i=1}^{n} f_i$, whereas our algorithm maintains separate piecewise linear models for each $f_i$. Furthermore, unlike our method, these methods require the workers to synchronize in every iteration.) Bundle methods have also been successfully applied to non-convex problems (see, for example, Hare & Sagastizábal (2010)). In particular, the special case of the Polyak step size, where the bundle only consists of the current cut and a lower bound on the objective, has proven to yield strong performance in deep neural network training (Loizou et al., 2021; Wang et al., 2023a).

## 3  ALGORITHM

In this section we present a model-based algorithm for solving (1) asynchronously. We use a parameter server architecture (Li et al., 2013) with one central server and $n$ workers. The central server maintains a copy of the global decision variable and can query each worker for its function value and gradient. Based on this information, the central server builds up a piecewise linear model of each worker's loss function, and then uses this model to compute the next iterate. A notable feature of the algorithm is that its implementation does not require knowledge of an upper bound on the information delay.

### 3.1  MAIN IDEA

To simplify the presentation of the method we consider a fixed iteration and drop the iteration index. We assume that the iteration number is sufficiently large to ensure that the central server has received information from each worker in at least $m$ previous iterates. The parameter $m$ is referred to as the *bundle size*. For $i \in \{1, \ldots, n\}$ we introduce an algorithmic parameter $M_i > 0$ which, roughly speaking, represents the smoothness parameter of worker $i$, and we let $M \triangleq \sum_{i=1}^{n} M_i$. (An exact definition of $M_i$ is given in §4.) We will later show that in a practical implementation of our method, the parameters $M_i$, $i = 1, \ldots, n$ are estimated adaptively and require no tuning.

Let $z_j^i$ for $j = 1, \ldots, m$ denote the $m$ previous iterates in which the central server has received information from worker $i$. We label the iterates so that $z_m^i$ is the *most recent* iterate for which the central server has received information from worker $i$, and let $\bar{z} \triangleq \frac{1}{M} \sum_{i=1}^{n} M_i z_m^i$ denote a weighted average of these points. The server maintains the following piecewise linear model of $f_i$:

$$\check{f}_i(x) = \max_{1 \leq j \leq m} \left\{ f_i(z_j^i) + \langle \nabla f_i(z_j^i), x - z_j^i \rangle \right\}. \tag{2}$$

When the server receives information from one or several workers, it replaces the oldest iterate in the bundle for those workers. The next iterate is then computed as an *approximate* solution of

$$\underset{x}{\text{minimize}} \left\{ \sum_{i=1}^{n} \check{f}_i(x) + R(x) + \frac{M}{2} \|x - \bar{z}\|_2^2 \right\}. \tag{3}$$

(We will later specify what we mean by an *approximate* solution.) Note in particular that the bundle center in (3) is chosen as a weighted average of the most recent iterates for the workers; this is essential for the convergence analysis in §4.

In our method, the central server must store $m$ gradients of size $d$ for all $n$ workers, resulting in a total memory complexity of order $\mathcal{O}(mnd)$. This is more than the $\mathcal{O}(nd)$ memory required by methods that only store the most recent gradient of each worker, but often substantially less than the $\mathcal{O}(d^2)$ requirement of methods such as (Soori et al., 2020) that store an approximation of the Hessian at the central server.

## 3.2 SOLVING THE MASTER PROBLEM

In every iteration the central server must solve the *master problem* (3), which for common regularizers such as $R(x) = \lambda\|x\|_1$ can be formulated as a quadratic program with linear inequality constraints. If the dimension $d$ is large, solving (3) can become a computational bottleneck. However, when an aggregated piecewise linear model of the objective function is used and $R(x) = 0$, it is well known that the dual of (3) is a low-dimensional quadratic program over the probability simplex (see, for example, (Hiriart-Urruty & Lemarechal, 1993, p. 296)). As the following lemma shows, a similar observation can be made when a disaggregated piecewise linear model of $f$ is used and when $R(x) \neq 0$. To state the lemma we define matrices $\boldsymbol{G}_i \in \mathbf{R}^{d \times m}$ containing old gradient information of $f_i$ by

$$\boldsymbol{G}_i = \begin{bmatrix} \nabla f_i(z_1^i) & \dots & \nabla f_i(z_m^i) \end{bmatrix}.$$

For $i = 1, \dots, n$, let $v_i \in \mathbf{R}^m$ be defined componentwise by $(v_i)_j = \langle \nabla f_i(z_j^i), z_j^i \rangle - f_i(z_j^i)$ and let $v = (v_1, \dots, v_n) \in \mathbf{R}^{mn}$. Recall that for $\gamma > 0$, the *Moreau envelope* of $R$ and the *proximal operator* of $R$ are defined by

$$H_R^\gamma(y) = \min_x \left\{ R(x) + \frac{1}{2\gamma}\|x - y\|_2^2 \right\}, \quad \mathbf{prox}_{\gamma R}(y) = \arg\min_x \left\{ R(x) + \frac{1}{2\gamma}\|x - y\|_2^2 \right\}.$$

**Lemma 3.1.** *Let* $\lambda_i \in \mathbf{R}^m$, $i = 1, \dots, n$ *and* $\lambda = (\lambda_1, \dots, \lambda_n) \in \mathbf{R}^{mn}$. *Define* $g : \mathbf{R}^{mn} \to \mathbf{R}$ *by*

$$g(\lambda) = \frac{M}{2}\|\bar{z} - \frac{1}{M}\sum_{i=1}^n \boldsymbol{G}_i\lambda_i\|_2^2 - H_R^{1/M}\left(\bar{z} - \frac{1}{M}\sum_{i=1}^n \boldsymbol{G}_i\lambda_i\right) + \langle v, \lambda \rangle.$$

*The Lagrange dual of (3) is given by*

$$\begin{array}{ll} minimize & g(\lambda) \\ subject\ to & \mathbf{1}^T\lambda_i = 1, \ \lambda_i \geq 0, \ i = 1, \dots, n. \end{array} \tag{4}$$

*Furthermore, if* $\lambda^\star$ *is optimal in (4), then the unique solution of (3), denoted by* $x_{exact}$, *is given by*

$$x_{exact} = \mathbf{prox}_{\frac{1}{M}R}\left(\bar{z} - \frac{1}{M}\sum_{i=1}^n \boldsymbol{G}_i\lambda_i^\star\right). \tag{5}$$

*Proof.* The proofs of this and all forthcoming results are given in the appendix. □

## 3.3 AN EFFICIENT APPROXIMATE MASTER PROBLEM SOLVER

According to Lemma 3.1, we can solve the master problem (3) by solving its low-dimensional dual (4). However, even if the dual is low-dimensional, it can be too expensive to solve it to high accuracy since the second term in the definition of the dual objective function itself involves a minimization problem in $x$. Therefore, in our algorithm, we only generate approximate solutions to (3) using inexact solutions to (4). The goal of this subsection is to introduce equation (6) below, which defines a termination criterion that we use to specify what we mean by an inexact solution.

Denote the feasible set of (4) by $\boldsymbol{\Delta} \subseteq \mathbf{R}^{mn}$. The dual objective function $g$ is differentiable since the Moreau envelope is differentiable. Hence, from optimality conditions for convex optimization (see, for example, (Nesterov, 2018, p. 177)), $\lambda^\star$ solves (4) if and only if

$$\langle \nabla g(\lambda^\star), \lambda^\star - \lambda \rangle \leq 0 \text{ for all } \lambda \in \boldsymbol{\Delta}.$$

---

**Algorithm 1**

---

**Setup**: $x_0$, parameters $M_i$, bundle size $m$, tolerance $\delta > 0$
**Initialization**: the central server receives $f_i(x_0)$ and $\nabla f_i(x_0)$, $i = 1, \ldots, n$
**while** not interrupted by central server: each worker $i$ **do**
    receive $x$ from the server, compute $f_i(x)$ and $\nabla f_i(x)$, and send them back to the server
**end while**
**while** not converged: central server **do**
    **for** $i = 1, \ldots, n$ **do**
        **if** received information from worker $i$ **then**
            update the bundle of worker $i$ by throwing out the oldest information
        **end if**
    **end for**
    compute $\bar{\lambda}$ satisfying (6) and then update $x$ according to (7)
    send back $x$ to all workers that the server received information from
**end while**

---

As in Nesterov & Florea (2021), we allow for inexact solution of (4) by introducing a parameter $\delta > 0$ together with the requirement that we compute a point $\bar{\lambda}$ satisfying

$$\langle \nabla g(\bar{\lambda}), \bar{\lambda} - \lambda \rangle \leq \delta \text{ for all } \lambda \in \mathbf{\Delta}. \tag{6}$$

The next iterate, denoted by $x_+$, is then computed as (cf. (5))

$$x_+ = \mathbf{prox}_{\frac{1}{M} R}\left( \bar{z} - \frac{1}{M} \sum_{i=1}^{n} \mathbf{G}_i \bar{\lambda}_i \right). \tag{7}$$

A summary of the algorithm we propose is given in Algorithm 1. Implementation details, including how to find $\bar{\lambda}$ satisfying (6), are given in §5.

### 3.4 EXTENSION TO STOCHASTIC FUNCTION VALUES AND GRADIENTS

While the main focus of this paper is the setting where exact (full batch) function and gradient evaluations are used, we will also analyze a variant that uses stochastic (mini-batch) function values and gradients. For this setting we assume that each worker $i \in \{1, \ldots, n\}$ has access to an oracle that when queried at a point $x$, draws a random variable $\xi$ from some distribution and outputs both a stochastic function value $F_i(x; \xi)$ approximating $f_i(x)$, and a stochastic gradient $G_i(x; \xi)$ approximating $\nabla f_i(x)$. In this stochastic setting, the central server replaces the piecewise linear model (2) of $f_i$ with the following stochastic piecewise linear model:

$$\check{f}_i(x, \boldsymbol{\xi}) = \max_{1 \leq j \leq m} \left\{ F_i(z_j^i; \xi_j^i) + \langle G_i(z_j^i; \xi_j^i), x - z_j^i \rangle \right\}.$$

Here $\xi_j^i$ is the random variable from the query of the oracle of worker $i$ in the point $z_j^i$, and the bold $\boldsymbol{\xi}$ represents all randomness used to construct the current bundle.

## 4 CONVERGENCE ANALYSIS

In this section we study the convergence of Algorithm 1 and its stochastic extension. Our main results are Theorem 4.5 and Theorem 4.9 showing that the algorithm converges to a neighborhood of an optimal solution whose size depends on the accuracy used to solve the master problem. They also characterize how the information delay affects the convergence rate. The convergence analysis uses two sequences of points: the sequence of iterates $x_k$ for $k \in \mathbf{Z}_+ \triangleq \{0, 1, 2, \ldots\}$ and the sequence of points $z_{k,j}^i$ for $k \in \mathbf{Z}_+, i \in \{1, \ldots, n\}$ and $j \in \{1, \ldots, m\}$ used to construct the piecewise linear model of $f_i$ in iteration $k$. (These points are previous iterates, *i.e.,* for $(k, i, j) \in \mathbf{Z}_+ \times \{1, \ldots, n\} \times \{1, \ldots, m\}$ there exists a non-negative integer $s_{k,j}^i \leq k$ such that $z_{k,j}^i = x_{s_{k,j}^i}$.)

We will assume that the points are labeled such that $z_{k,m}^i$ denotes the *most recent iterate* in which the central server has received information from worker $i$ in iteration $k$. With this convention, the

quantity $k - s_{k,m}^i \geq 0$ is called the *delay of worker $i$ in iteration $k$*. Our first assumption is standard (see, for example, Feyzmahdavian et al. (2014); Gürbüzbalaban et al. (2017); Vanli et al. (2018); Mishchenko et al. (2018)) and states that the maximum delay is bounded by an integer $\tau \geq 0$.

**Assumption 4.1.** In every iteration $k$ the delay of worker $i$ is bounded by $\tau$. In other words, for $(k, i) \in \mathbf{Z}_+ \times \{1, \ldots, n\}$ it holds that $k - s_{k,m}^i \leq \tau$.

We will also make the following standard assumption on the objective function.

**Assumption 4.2.** The loss function of worker $i \in \{1, \ldots, n\}$ is smooth with parameter $L_i$.

Occasionally we will further make the following common growth assumption that is similar to, but weaker, than strong convexity (see, for example, Necoara et al. (2019)).

**Assumption 4.3.** The full objective function $F$ has a quadratic functional growth with parameter $\mu > 0$, meaning that $F(x) - F(x^\star) \geq (\mu/2)\|x - x^\star\|_2^2$ for all $x \in \mathbf{R}^d$.

### 4.1 ANALYSIS FOR EXACT FUNCTION VALUES AND GRADIENTS

We now present a convergence analysis when exact (non-stochastic) function values and gradients are used. First we need an additional assumption.

**Assumption 4.4.** The loss function $f_i$ of worker $i \in \{1, \ldots, n\}$ is star-convex, meaning that $f_i(x^\star) \geq f_i(x) + \langle \nabla f_i(x), x^\star - x \rangle$ for all $x \in \mathbf{R}^d$.

Under the growth assumption we can show linear convergence to a neighborhood of the solution.

**Theorem 4.5.** *Under Assumptions 4.1, 4.2, 4.3 and 4.4 the iterates of Algorithm 1 using $M_i = L_i$, $i = 1, \ldots, n$ satisfy*

$$\|x_k - x^\star\|_2^2 \leq \rho^k \|x_0 - x^\star\|_2^2 + \epsilon_\delta, \tag{8}$$

*where $\rho = (L/(L + \mu))^{1/(1+\tau)}$ and $\epsilon_\delta = 2\delta/\mu$.*

*Remark* 4.6. A notable feature of our algorithm is that neither its implementation nor its tuning requires any information about the level of asynchrony in the system. The method converges with default parameters as long as the information delay from all workers are finite, and under Assumption 4.3, the convergence rate decreases as the level of asynchrony in the system increases.

*Remark* 4.7. Rather than relying on the common yet unrealistic assumption of solving the subproblem (3) exactly, our analysis explicitly accounts for and characterizes the impact of inexact subproblem solutions. As a result, an error term depending on $\delta$ naturally appears in the convergence result. However, we should point out that in practice, the algorithm we propose has no issues with finding highly accurate solutions (see §6).

For the analysis without the growth assumption we will use the following new sequence result that might be of independent interest.

**Lemma 4.8.** *Suppose that $(V_k)_{k=0}^\infty$ and $(W_k)_{k=0}^\infty$ are non-negative sequences satisfying*

$$V_{k+1} \leq \max_{(k-\tau)_+ \leq \ell \leq k} V_\ell - W_{k+1} + r, \qquad k = 0, 1, 2,$$

*for a non-negative constant $r$. Then, for any $k \geq 1$,*

$$\min_{t \leq k} W_t \leq \frac{(\tau + 1)V_0}{k} + r.$$

Using Lemma 4.8 we can prove sublinear convergence in terms of the function value gap.

**Theorem 4.9.** *Under Assumptions 4.1, 4.2, and 4.4, the iterates of Algorithm 1 using $M_i = L_i$, $i = 1, \ldots, n$, satisfy that for any $k \geq 1$,*

$$\min_{t \leq k} F(x_t) - F(x^\star) \leq \frac{(\tau + 1)L\|x_0 - x^\star\|_2^2}{2k} + \delta.$$

Table 1 compares the convergence rates of our method (with $\delta = 0$) against two asynchronous proximal gradient methods: `DAve-RPG` (Mishchenko et al., 2018; 2020) and `PIAG` with delay-tracking

(Wu et al., 2022), under the assumptions used in this paper. These methods operate under the same conditions as ours in that they rely on a parameter server, do not require explicit maximum delay information in their parameters, and explicitly incorporate the regularizer. The comparison shows that all three methods achieve similar convergence rates under standard convexity assumptions. However, in the strong convexity-like setting, DAve-RPG exhibits the fastest rate. This is expected as our method is more complex, making the analysis more challenging and leading to a looser bound.

| Assumptions | DAve-RPG | Delay-tracking PIAG | Algorithm 1 |
|---|---|---|---|
| General convexity | $\min_{t \leq k} \|g_t\|^2 \leq \mathcal{O}(1/k)$ | $F(x_k) - F(x^\star) \leq \mathcal{O}(1/k)$ | $\min_{t \leq k} F(x_t) - F(x^\star) \leq \mathcal{O}(1/k)$ |
| Strong convexity-like | $\|x_k - x^\star\|_2^2 \leq \rho_1^k \|x_0 - x^\star\|_2^2$ | $F(x_k) - F(x^\star) \leq \rho_2^k (F(x_0) - F(x^\star))$ | $\|x_k - x^\star\|_2^2 \leq \rho_3^k \|x_0 - x^\star\|_2^2$ |

Table 1: Comparison of convergence guarantees under different assumptions. Here $\rho_1 = ((L - \mu)/(L + \mu))^{2/(\tau+1)}$, $\rho_2 = e^{-\mu/(9L(\tau+1))} \approx 1 - \frac{\mu}{9L(\tau+1)}$, and $\rho_3 = (L/(L + \mu))^{1/(\tau+1)}$.

## 4.2 ANALYSIS FOR STOCHASTIC FUNCTION VALUES AND GRADIENTS

When stochastic function values and gradients are used we will make the following assumptions.

**Assumption 4.10.** For each worker $i \in \{1, \ldots, n\}$:

1. The oracle is star-convex, *i.e.*, $F_i(x^\star; \xi) \geq F_i(x; \xi) + \langle G_i(x; \xi), x^\star - x \rangle$ for all $x \in \mathbf{R}^d$. Furthermore, $\mathbf{E}[F_i(x, \xi)] = f_i(x)$ and $\mathbf{E}[G_i(x, \xi)] = \nabla f_i(x)$ for all $x \in \mathbf{R}^d$.

2. The variance of the stochastic gradients is bounded by some finite constant $\sigma_2^2 > 0$, meaning that $\mathbf{E}[\|G_i(x, \xi) - \nabla f_i(x)\|_2^2] \leq \sigma_2^2$ for all $x \in \mathbf{R}^d$.

3. The function value noise at the optimal solution $x^\star$ is bounded by some finite constant $\sigma_1^2 > 0$, meaning that $\mathbf{E}[(F_i(x^\star; \xi) - f_i(x^\star))^2] \leq \sigma_1^2$.

*Relation to previous assumptions in the literature.* The second assumption bounding the noise of the gradients is common in the analysis of stochastic algorithms (see, for example, Koloskova et al. (2022); Mishchenko et al. (2022)). The third assumption is less common, since most algorithms often only use stochastic gradients and *not* function values. However, recent analysis of stochastic algorithms that use stochastic function values in addition to stochastic gradients make a similar assumption (see, for example, Loizou et al. (2021); Wang et al. (2023b)).

**Theorem 4.11.** *Consider Algorithm 1 with $M_i = \alpha L_i$, $i = 1, \ldots, n$ where $\alpha > 1$. Assume that stochastic function values and gradients are used. Under Assumptions 4.1, 4.2, and 4.10, the iterates of Algorithm 1 satisfy that for any $k \geq 1$,*

$$\min_{t \leq k} \mathbf{E}[F(x_t)] - F(x^\star) \leq \frac{\alpha(\tau + 1)L\|x - x_0\|_2^2}{2k} + \epsilon, \tag{9}$$

*where $\epsilon = \epsilon_\delta + \epsilon_{\sigma_1} + \epsilon_{\sigma_2}$ with*

$$\epsilon_\delta = \delta, \qquad \epsilon_{\sigma_1} = n\sigma_1\sqrt{m}, \qquad \epsilon_{\sigma_2} = \frac{\sigma_2^2}{2(\alpha - 1)} \cdot \sum_{i=1}^n \frac{1}{L_i}.$$

*If, in addition, Assumption 4.3 holds, then*

$$\mathbf{E}[\|x_k - x^\star\|_2^2] \leq \rho^k \|x_0 - x^\star\|_2^2 + 2\epsilon/\mu,$$

*where $\rho = (\alpha L/(\alpha L + \mu))^{1/(1+\tau)}$.*

*Remark* 4.12. Compared to algorithms with a simple explicit update rule of the form $x_{k+1} = \mathbf{prox}_{\gamma g}(x_k + \gamma d_k)$ where $d_k$ is a direction and $\gamma$ is a step size, the update mechanism of Algorithm 1 is more implicit since it involves solving the dual subproblem (4) approximately. This makes the analysis challenging. One of the main technical challenges in the proof in the stochastic setting is to carefully manage correlations between recently queried gradients $G_i(z_m^i)$, the dual variable $\tilde{\lambda}$, and the next iterate $x_{k+1}$, all of which are correlated random variables.

*Remark* 4.13. The size of the neighborhood of the solution that $x_k$ converges to in expectation depends on three terms: one term $\epsilon_\delta$ which depends on the accuracy $\delta$, and two other terms $\epsilon_{\sigma_1}$ and $\epsilon_{\sigma_2}$ which depend on the strength of the noise. The noise terms $\epsilon_{\sigma_1}$ and $\epsilon_{\sigma_2}$ depend on $n$, which may seem uncommon. This dependency arises because we analyze the canonical form $f(x) = \sum_{i=1}^n f_i(x)$ instead of the more common form $f(x) = (1/n)\sum_{i=1}^n f_i(x)$. Under the latter form, the noise terms would not depend on $n$.

## 5 Implementation

When implementing Algorithm 1, two issues must be addressed.

*Solving the subproblem.* First, we must find an approximate solution to (4) by finding $\bar\lambda \in \mathbf{R}^{mn}$ satisfying (6). Since $\mathbf{\Delta}$ is a Cartesian product of simplices, it is cheap to verify condition (6) by noting that

$$\sup_{\lambda \in \mathbf{\Delta}} \langle \nabla g(\bar\lambda), \bar\lambda - \lambda \rangle = \langle \nabla g(\bar\lambda), \bar\lambda \rangle - \inf_{\lambda \in \mathbf{\Delta}} \langle \nabla g(\bar\lambda), \lambda \rangle = \langle \nabla g(\bar\lambda), \bar\lambda \rangle - \sum_{i=1}^n \min \nabla_i g(\bar\lambda),$$

where $\min \nabla_i g(\bar\lambda)$ is the smallest element of $\nabla_i g(\bar\lambda)$. (Here $\nabla_i g(\bar\lambda)$ is the gradient of $g$ with respect to $\lambda_i$.) To find $\bar\lambda$ we have implemented an accelerated projected gradient method for solving (4) (Beck, 2017, page 291). Each iteration requires the gradient of $g$ and projecting onto the feasible set $\mathbf{\Delta}$. Since $\mathbf{\Delta}$ is a Cartesian product of low-dimensional simplices, the projection can be done efficiently (Condat, 2016). Furthermore, from properties of the Moreau-envelope (Beck, 2017, p. 166) it follows that the gradient of $g$ with respect to $\lambda_i$ is

$$\nabla_i g(\lambda) = \boldsymbol{G}_i^T(u - \mathbf{prox}_{\frac{1}{M}R}(u)) - \boldsymbol{G}_i^T u + v_i = v_i - \boldsymbol{G}_i^T \mathbf{prox}_{\frac{1}{M}R}(u), \tag{10}$$

where $u \triangleq \bar z - \frac{1}{M}\sum_{i=1}^n \boldsymbol{G}_i \lambda_i$.

In the appendix we compare the cost of solving the subproblem using this specialized method versus a high-performance interior-point solver. The main conclusion is that this specialized approach is more than an order of magnitude faster and that the complexity for solving the subproblem is of order $\mathcal{O}(nmd)$.

*Adaptive estimation of smoothness parameters.* First-order methods for solving (1) typically require knowledge of smoothness parameters. These parameters are often unknown or expensive to compute in practice. To eliminate the need for choosing a suitable value on $L_i$ in Algorithm 1 we propose to estimate it adaptively using similar ideas to Malitsky & Mishchenko (2020).

Recall that $\nabla f_i(z_m^i)$ is the *most recent* gradient that the central server has received from worker $i$, and let $\nabla f_i(z_{m-1}^i)$ denote the *next most recent* gradient. Given $z_m^i$, $\nabla f_i(z_m^i)$, $z_{m-1}^i$ and $\nabla f_i(z_{m-1}^i)$, a natural estimate of the local smoothness of $f_i$ is the quantity $\hat L_i = \|\nabla f_i(z_m^i) - \nabla f_i(z_{m-1}^i)\|_2 / \|z_m^i - z_{m-1}^i\|_2$. Every time the central server receives a new gradient from worker $i$, we propose to update the smoothness parameter $L_i$ using this estimate.

## 6 Experiments

We consider binary and multiclass classification problems based on a logistic model. For the binary classification, we use the objective function $f(x) = (1/N)\sum_{j=1}^N \left( \log(1 + e^{-y_j(a_j^T x)}) + \frac{\lambda_2}{2}\|x\|_2^2 \right)$ and the regularizer $R(x) = \lambda_1\|x\|_1$, where $a_1, \ldots, a_N \in \mathbf{R}^p$ are the feature vectors and $y_1, \ldots, y_N \in \{-1, 1\}$ are the corresponding labels. Due to the space limitations, we defer the results for multiclass classification to the appendix.

We conduct experiments on three datasets (`mnist8m`/`infimnist`, `epsilon`, `rcv1`) from the LIBSVM library (Chang & Lin, 2011) and on the `SVHN` dataset (Netzer et al., 2011). We pick $\lambda_2 = 1/N$, and tune $\lambda_1$ for each dataset to obtain a classifier $x^\star$ with 10-20% non-zero entries. The dataset `mnist8m` corresponds to a multiclass problem with 10 different labels. To use it for binary classification, we select data corresponding to the digits 7 and 9 and discard the rest. Table 2 in Appendix A.4 shows the dimensions of each problem and the value of $\lambda_1$.

All methods are evaluated on a workstation using 10 cores. One core is assigned the role as the central server and the remaining $n = 9$ cores are workers. The data is distributed evenly among the workers. The code is written in Python using MPI4PY (Dalcin & Fang, 2021) and is available at `https://github.com/dance858/Asynchronous-bundle-method`. To evaluate the objective value and the gradients we use PyTorch (Paszke et al., 2019) for the dense datasets, and sparse linear algebra for `rcv1`.

## 6.1 BENCHMARKING

To benchmark our asynchronous bundle method (`ABM`) we compare it with two asynchronous proximal gradient methods, namely `DAve-RPG` (Mishchenko et al., 2018) and `PIAG` with delay-tracking (Wu et al., 2022). We selected these methods since they operate under the same conditions as `ABM` in the sense that they use a parameter server, require no maximum delay information in parameters, and explicitly incorporate the regularizer. Both `DAve-RPG` and `PIAG` use exact gradients so we also use exact function values and gradients for `ABM`.

For `ABM` we use bundle size $m = 10$, master problem tolerance $\delta = 10^{-7}$, and adaptive smoothness estimation. `DAve-RPG` has two hyperparameters: the step size $\gamma$ and the number of inner prox-steps $p$. We use step size $\gamma = 1/L_{\text{average}}$ where $L_{\text{average}}$ is the average smoothness parameter of the workers, and $p = 1$ inner prox-steps (as in Mishchenko et al. (2018)). For `PIAG` with delay-tracking we implemented the first adaptive step size strategy described in Wu et al. (2022).

The first row of Figure 1 shows the relative suboptimality $(f(x_k) - f^\star)/f^\star$ versus the runtime. We see that `ABM` clearly outperforms the other two methods. The second row shows the suboptimality versus the number of gradients received by the server. We see that `ABM` achieves much higher accuracy with fewer gradients received by the server. For `rcv1` the difference is striking: in 300 seconds the central server in `ABM` receives about 4000 gradients, while the servers of `DAve-RPG` and `PIAG` receive more than 70000 gradients in the same amount of time. This is in contrast to `mnist8m` and `epsilon` where all servers receive roughly the same number of gradients in the same amount of time. The reason underlying this observation is that the gradients are cheap to compute for the sparse data set `rcv1` and more expensive to compute for the dense datasets (cf. Table 2 in Appendix A.4). Consequently, for `rcv1`, the time required to solve the subproblem at the central server is non-negligible compared to the time needed to evaluate gradients. In contrast, for the dense datasets, this computation time is almost negligible.

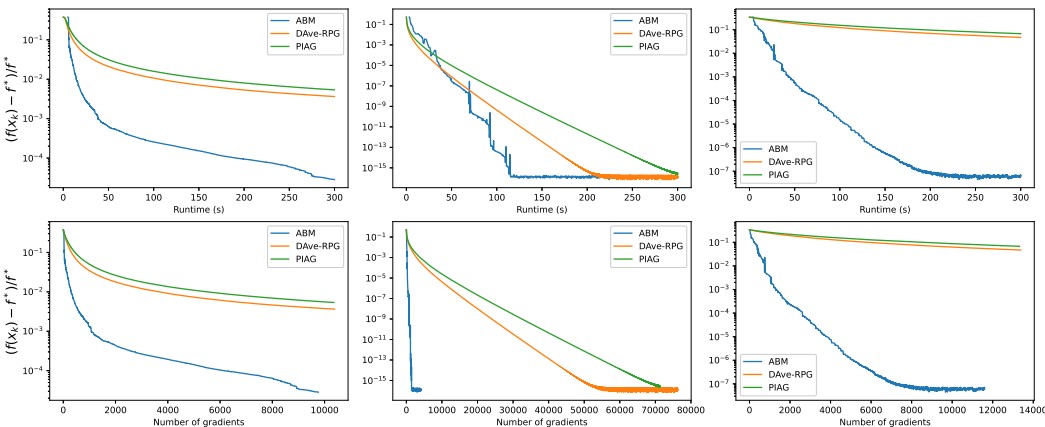

Figure 1: The progress of `ABM`, `DAve-RPG` and `PIAG` on the binary classification problems. The datasets are arranged in the order `mnist8m`, `rcv1` and `epsilon` from the left.

## 6.2 SENSITIVITY TO HYPERPARAMETERS

Strictly speaking, `ABM` has two hyperparameters: the tolerance $\delta$ and the bundle size $m$. We will now investigate the sensitivity of the algorithm's performance to these parameters. The first row of Figure 2 shows the progress of `ABM` for fixed $\delta = 10^{-7}$ and bundle size $m \in \{2, 5, 10\}$. Increasing

the bundle size from $m = 2$ to $m \in \{5, 10\}$ results in much faster convergence for mnist8m. Furthermore, ABM makes *no* progress with bundle size $m = 2$ for rcv1 and epsilon, but with bundle size $m \in \{5, 10\}$ the convergence is fast. This indicates the advantage of using a more accurate approximation of the objective function for computing the next iterate. Next, we run ABM with fixed bundle size $m = 10$ and tolerance $\delta \in \{10^{-5}, 10^{-7}, 10^{-9}\}$. The result is shown in the second row of Figure 2. We see that ABM has good performance for all three values on $\delta$. This experiment suggests that ABM essentially requires no tuning in these experiments: the values $m = 10$ and $\delta = 10^{-7}$ seem to work well. In particular, the performance of ABM is not as sensitive to its hyperparameters as, for example, stochastic gradient descent is to its step size.

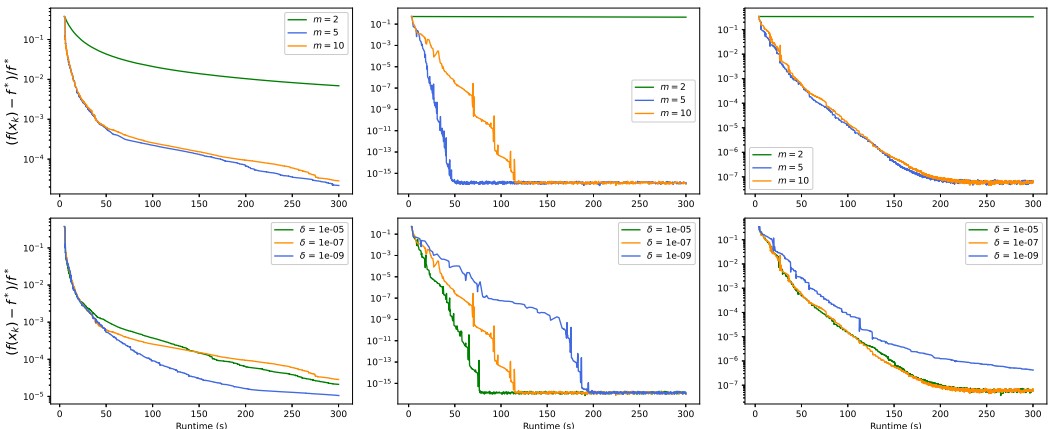

Figure 2: Sensitivity of ABM's performance with respect to hyperparameters. The first row shows the performance for different bundle sizes $m$ and fixed tolerance $\delta$. The second row shows the performance for different tolerances $\delta$ and fixed bundle size. Each column corresponds to one dataset, and the columns are arranged in the order mnist8m, rcv1, epsilon.

### 6.3 FURTHER EXPERIMENTS

In the appendix we conduct further experiments when $\lambda_2 = 0$, *i.e.,* when the objective function is convex but not strongly convex, and we also conduct experiments on multiclass classification. The main conclusion is that ABM outperforms DAve-RPG and PIAG with delay tracking (see Figure 3 and 4 in the appendix). We also test the stochastic version of ABM using mini-batches for multiclassification on the biggest data set mnist8m. The experiment indicates that mini-batching can speed up ABM if only a modest accuracy is required (see Figure 5 in the appendix).

## 7 DISCUSSION

We have presented an asynchronous bundle method that is suitable for distributed learning problems. The algorithm constructs a piecewise linear model to approximate the local loss of each worker and uses this model to compute the next iterate. Compared to other first-order asynchronous algorithms, our proposed method employs a more refined model of the objective function. This allows it to converge quickly in practice with minimal tuning or specification of unknown constants.

Our method uses a fixed bundle size. An interesting extension would be to design a scheme that dynamically adjusts the bundle size based on the actual delays, potentially discarding outdated function value information that hasn't been used recently in the server subproblem. (The $j$th linear approximation of $f_i$ is not used in the server subproblem if $\lambda_{ij}^\star = 0$, where $\lambda^\star$ is the solution of (4).) Another extension could be to let the central server maintain a low-rank approximation of the Hessian for each worker, enabling a better approximation of the curvature of the loss function, and possibly faster convergence. We leave these extensions for future work.

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

## A  APPENDIX

In this appendix we present proofs and additional numerical experiments.

### A.1  DUAL SUBPROBLEM

**Lemma A.1.** *Let $\lambda_i \in \mathbf{R}^m$, $i = 1, \ldots, n$ and $\lambda = (\lambda_1, \ldots, \lambda_n) \in \mathbf{R}^{mn}$. Define $g : \mathbf{R}^{mn} \to \mathbf{R}$ by*

$$g(\lambda) = \frac{M}{2} \|\bar{z} - \frac{1}{M} \sum_{i=1}^{n} \boldsymbol{G}_i \lambda_i\|_2^2 - H_R^{1/M}\left(\bar{z} - \frac{1}{M} \sum_{i=1}^{n} \boldsymbol{G}_i \lambda_i\right) + \langle v, \lambda \rangle.$$

*The Lagrange dual of (3) is given by*

$$\begin{aligned} &\text{minimize} \quad g(\lambda) \\ &\text{subject to} \quad \mathbf{1}^T \lambda_i = 1, \ \lambda_i \geq 0, \ i = 1, \ldots, n. \end{aligned} \tag{11}$$

*Furthermore, if $\lambda^\star$ is optimal in (4), then the unique solution of (3), denoted by $x_{exact}$, is given by*

$$x_{exact} = \mathbf{prox}_{\frac{1}{M}R}\left(\bar{z} - \frac{1}{M} \sum_{i=1}^{n} \boldsymbol{G}_i \lambda_i^\star\right). \tag{12}$$

*Proof.* In the proof we use the notation of stochastic function values and gradients. The setting with exact function evaluations can be recovered by doing the substitutions $F_i(x; \xi) = f_i(x)$ and $G_i(x; \xi) = \nabla f_i(x)$.

Problem (3) can be formulated as

$$\begin{aligned} &\text{minimize} \sum_{i=1}^{n} r_i + \frac{M}{2} \|x - \bar{z}\|_2^2 + R(x) \\ &\text{subject to} \ r_i \geq F_i(z_j^i; \xi_j^i) + \langle G_i(z_j^i; \xi_j^i), x - z_j^i \rangle, \ j = 1, \ldots, m, \ i = 1, \ldots, n, \end{aligned} \tag{13}$$

with variables $r \in \mathbf{R}^n$ and $x \in \mathbf{R}^d$. For $i = 1, \ldots, n$, introduce a Lagrange multiplier vector $\lambda_i \in \mathbf{R}^m$. The Lagrangian is given by

$$\begin{aligned} \mathcal{L}(x, r, \lambda_1, \ldots, \lambda_n) &= \sum_{i=1}^{n} r_i + \frac{M}{2} \|x - \bar{z}\|_2^2 + \sum_{i=1}^{n} \sum_{j=1}^{m} \lambda_{ij}(F_i(z_j^i; \xi_j^i) + \langle G_i(z_j^i; \xi_j^i), x - z_j^i \rangle - r_i) + R(x) \\ &= \sum_{i=1}^{n} \left(1 - \mathbf{1}^T \lambda_i\right) r_i + \frac{M}{2} \|x - \bar{z}\|_2^2 + \sum_{i=1}^{n} \langle \lambda_i, \boldsymbol{G}_i^T x - v_i \rangle + R(x), \end{aligned}$$

where

$$\boldsymbol{G}_i = \begin{bmatrix} G_i(z_1^i; \xi_1^i) & \ldots & G_i(z_m^i; \xi_m^i) \end{bmatrix} \in \mathbf{R}^{d \times m}$$

and $v_i \in \mathbf{R}^m$ is defined componentwise by

$$(v_i)_j = \langle G_i(z_j^i; \xi_j^i), z_j^i \rangle - F_i(z_j^i; \xi_j^i).$$

The Lagrangian is unbounded in $r$ unless $\mathbf{1}^T \lambda_i = 1$, $i = 1, \ldots, n$. Furthermore, for such $\lambda$ minimizing the Lagrangian over $x$ yields

$$\inf_x \mathcal{L}(x, r, \lambda_1, \ldots, \lambda_n) = \inf_x \left\{ R(x) + \frac{M}{2} \|x - \bar{z}\|_2^2 + \sum_{i=1}^{n} \langle \boldsymbol{G}_i \lambda_i, x \rangle \right\} - \sum_{i=1}^{n} \langle \lambda_i, v_i \rangle$$

$$= \inf_x \left\{ R(x) + \frac{M}{2} \|x - (\bar{z} - \frac{1}{M} \sum_{i=1}^{n} \boldsymbol{G}_i \lambda_i)\|_2^2 \right\} - \frac{M}{2} \|\bar{z} - \frac{1}{M} \sum_{i=1}^{n} \boldsymbol{G}_i \lambda_i\|_2^2 + \frac{M}{2} \|\bar{z}\|_2^2 - \sum_{i=1}^{n} \langle \lambda_i, v_i \rangle$$

$$= H_R^{1/M}\left(\bar{z} - \frac{1}{M} \sum_{i=1}^{n} \boldsymbol{G}_i \lambda_i\right) - \frac{M}{2} \|\bar{z} - \frac{1}{M} \sum_{i=1}^{n} \boldsymbol{G}_i \lambda_i\|_2^2 + \frac{M}{2} \|\bar{z}\|_2^2 - \sum_{i=1}^{n} \langle \lambda_i, v_i \rangle.$$

By dropping the term $\frac{M}{2}\|\bar{z}\|_2^2$ it follows that a dual problem is given by

$$\text{minimize } g(\lambda) := -H_R^{1/M}\left(\bar{z} - \frac{1}{M}\sum_{i=1}^n \boldsymbol{G}_i \lambda_i\right) + \frac{M}{2}\|\bar{z} - \frac{1}{M}\sum_{i=1}^n \boldsymbol{G}_i \lambda_i\|_2^2 + \sum_{i=1}^n \langle \lambda_i, v_i \rangle$$

$$\text{subject to } \lambda \in \boldsymbol{\Delta}.$$

Since Slater's constraint qualification (Boyd & Vandenberghe, 2004, page 226) is satisfied, strong duality holds and the point $x_{\text{exact}}$ defined by (12) minimizes $\mathcal{L}(x, r, \lambda_1^\star, \ldots, \lambda_n^\star)$ over $x$ (here the value of $r$ is arbitrary since the Lagrangian is independent of $r$ for any value of $\lambda$ that is dual feasible). The function $x \mapsto \mathcal{L}(x, r, \lambda_1^\star, \ldots, \lambda_n^\star)$ has a unique minimizer, so it follows that $x_{\text{exact}}$ given by (12) indeed solves (3). $\qquad\square$

## A.2 CONVERGENCE ANALYSIS

In the analysis below we analyze the progress Algorithm 1 makes in iteration $k$. To simplify the notation we drop the iteration index. In other words, the notation $z_j^i$, $j = 1, \ldots, m$ below refers to the points used to construct the piecewise linear model of $f_i$ in iteration $k$.

In the first few results we will stick with the convention of using stochastic function values and gradients. The setting with exact function evaluations can be recovered by doing the substitutions $F_i(x; \xi) = f_i(x)$ and $G_i(x; \xi) = \nabla f_i(x)$.

We will analyze Algorithm 1 by using (7). This will make the analysis depend on the dual variable $\bar{\lambda}$. The following result (inspired by Nesterov & Florea (2021)) will be useful to partly remove the dependence on $\bar{\lambda}$ from the analysis.

**Lemma A.2.** *Assume $\bar{\lambda}$ satisfies (6) and let $x_{k+1}$ be given by (7). Then*

$$\sum_{i=1}^n \sum_{j=1}^m \bar{\lambda}_{ij}[F_i(z_j^i; \xi_j^i) + \langle G_i(z_j^i; \xi_j^i), x_{k+1} - z_j^i \rangle] \geq \sum_{i=1}^n \check{f}_i(x_{k+1}; \boldsymbol{\xi}) - \delta.$$

*Proof.* Since (6) is satisfied we have $\langle \bar{\lambda}, -\nabla g(\bar{\lambda}) \rangle \geq \sup_{\lambda \in \boldsymbol{\Delta}} \langle \lambda, -\nabla g(\bar{\lambda}) \rangle - \delta$. Note that

$$\langle \bar{\lambda}, -\nabla g(\bar{\lambda}) \rangle = -\sum_{i=1}^n \langle \bar{\lambda}_i, \nabla g_i(\bar{\lambda}) \rangle = \sum_{i=1}^n \sum_{j=1}^m \bar{\lambda}_{ij}[F_i(z_j^i; \xi_j^i) + \langle G_i(z_j^i; \xi_j^i), x_{k+1} - z_j^i \rangle]$$

$$\sup_{\lambda \in \boldsymbol{\Delta}} \langle \lambda, -\nabla g(\bar{\lambda}) \rangle = \sup_{\lambda \in \boldsymbol{\Delta}} \sum_{i=1}^n \sum_{j=1}^m \lambda_{ij}[F_i(z_j^i; \xi_j^i) + \langle G_i(z_j^i; \xi_j^i), x_{k+1} - z_j^i \rangle] = \sum_{i=1}^n \check{f}_i(x_{k+1}, \boldsymbol{\xi}).$$

$\qquad\square$

**Lemma A.3.** *The next iterate $x_{k+1}$ satisfies*

$$\frac{1}{2}\|x_{k+1} - x^\star\|_2^2 - \frac{1}{2M}\sum_{i=1}^n M_i \|z_m^i - x^\star\|_2^2 \leq \frac{1}{M}(R(x^\star) - R(x_{k+1})) - \frac{1}{2M}\sum_{i=1}^n M_i \|x_{k+1} - z_m^i\|_2^2$$

$$+ \frac{1}{M}\sum_{i=1}^n \sum_{j=1}^m \bar{\lambda}_{ij} \langle G_i(z_j^i; \xi_j^i), x^\star - x_{k+1} \rangle. \tag{14}$$

*Proof.* In the proof we will apply the identity

$$\sum_{i=1}^n \frac{M_i}{2}\|y - z_m^i\|_2^2 = \frac{M}{2}\|y - \frac{1}{M}\sum_{i=1}^n M_i z_m^i\|_2^2 - \frac{1}{2M}\|\sum_{i=1}^n M_i z_m^i\|_2^2 + \sum_{i=1}^n \frac{M_i}{2}\|z_m^i\|_2^2$$

twice; once with $y = x^\star$ and once with $y = x_{k+1}$.

Using the three-points lemma $\frac{1}{2}\|b - c\|_2^2 - \frac{1}{2}\|a - c\|_2^2 = \langle a - b, c - b \rangle - \frac{1}{2}\|a - b\|_2^2$ we get

$$\frac{1}{2}\|x_{k+1} - x^\star\|_2^2 - \frac{1}{2}\|\bar{z} - x^\star\|_2^2 = \langle \bar{z} - x_{k+1}, x^\star - x_{k+1} \rangle - \frac{1}{2}\|x_{k+1} - \bar{z}\|_2^2. \tag{15}$$

Since $x_{k+1} = \mathbf{prox}_{\frac{1}{M}R}(\bar{z} - \frac{1}{M}\sum_{i=1}^{n}\sum_{j=1}^{m}\bar{\lambda}_{ij}G_i(z_j^i;\xi_j^i))$ it follows from optimality conditions for convex optimization that (Nesterov, 2018, Thm 3.1.23)

$$R(x_{k+1}) \leq R(y) + M\langle x_{k+1} - (\bar{z} - \frac{1}{M}\sum_{i=1}^{n}\sum_{j=1}^{m}\bar{\lambda}_{ij}G_i(z_j^i;\xi_j^i)), y - x_{k+1}\rangle \text{ for all } y \in \mathbb{R}^d.$$

If we let $y = x^\star$ and rearrange we get

$$\langle \bar{z} - x_{k+1}, x^\star - x_{k+1}\rangle \leq \frac{1}{M}(R(x^\star) - R(x_{k+1})) + \frac{1}{M}\sum_{i=1}^{n}\sum_{j=1}^{m}\bar{\lambda}_{ij}\langle G_i(z_j^i;\xi_j^i), x^\star - x_{k+1}\rangle.$$

Using this bound in (15) shows that

$$\frac{1}{2}\|x_{k+1} - x^\star\|_2^2 - \frac{1}{2}\|\bar{z} - x^\star\|_2^2 \leq \frac{1}{M}(R(x^\star) - R(x_{k+1})) + \frac{1}{M}\sum_{i=1}^{n}\sum_{j=1}^{m}\bar{\lambda}_{ij}\langle G_i(z_j^i;\xi_j^i), x^\star - x_{k+1}\rangle$$

$$- \frac{1}{2}\|x_{k+1} - \bar{z}\|_2^2$$

$$= \frac{1}{M}(R(x^\star) - R(x_{k+1})) + \frac{1}{M}\sum_{i=1}^{n}\sum_{j=1}^{m}\bar{\lambda}_{ij}\langle G_i(z_j^i;\xi_j^i), x^\star - x_{k+1}\rangle$$

$$- \frac{1}{M}\left(\sum_{i=1}^{n}\frac{M_i}{2}\|x_{k+1} - z_m^i\|_2^2 + \frac{1}{2M}\|\sum_{i=1}^{n}M_i z_m^i\|_2^2 - \sum_{i=1}^{n}\frac{M_i}{2}\|z_m^i\|_2^2\right).$$

We can rearrange to obtain

$$\frac{1}{2}\|x_{k+1} - x^\star\|_2^2 - \frac{1}{2}\left(\|\frac{1}{M}\sum_{i=1}^{n}M_i z_m^i - x^\star\|_2^2 - \frac{1}{M^2}\|\sum_{i=1}^{n}M_i z_m^i\|_2^2 + \frac{1}{M}\sum_{i=1}^{n}M_i\|z_m^i\|_2^2\right)$$

$$\leq \frac{1}{M}(R(x^\star) - R(x_{k+1})) + \frac{1}{M}\sum_{i=1}^{n}\sum_{j=1}^{m}\bar{\lambda}_{ij}\langle G_i(z_j^i;\xi_j^i), x^\star - x_{k+1}\rangle - \frac{1}{2M}\sum_{i=1}^{n}M_i\|x_{k+1} - z_m^i\|_2^2.$$

Note that

$$\|\frac{1}{M}\sum_{i=1}^{n}M_i z_m^i - x^\star\|_2^2 - \frac{1}{M^2}\|\sum_{i=1}^{n}M_i z_m^i\|_2^2 + \frac{1}{M}\sum_{i=1}^{n}M_i\|z_m^i\|_2^2$$

$$= \frac{2}{M}\left(\frac{M}{2}\|x^\star - \frac{1}{M}\sum_{i=1}^{n}M_i z_m^i\|_2^2 - \frac{1}{2M}\|\sum_{i=1}^{n}M_i z_m^i\|_2^2 + \sum_{i=1}^{n}\frac{M_i}{2}\|z_m^i\|_2^2\right)$$

$$= \frac{2}{M}\sum_{i=1}^{n}\frac{M_i}{2}\|x^\star - z_m^i\|_2^2 = \frac{1}{M}\sum_{i=1}^{n}M_i\|x^\star - z_m^i\|_2^2.$$

Hence, we conclude that

$$\frac{1}{2}\|x_{k+1} - x^\star\|_2^2 - \frac{1}{2M}\sum_{i=1}^{n}M_i\|z_m^i - x^\star\|_2^2 \leq \frac{1}{M}(R(x^\star) - R(x_{k+1})) - \frac{1}{2M}\sum_{i=1}^{n}M_i\|x_{k+1} - z_m^i\|_2^2$$

$$+ \frac{1}{M}\sum_{i=1}^{n}\sum_{j=1}^{m}\bar{\lambda}_{ij}\langle G_i(z_j^i;\xi_j^i), x^\star - x_{k+1}\rangle.$$

$\square$

**Lemma A.4.** *The next iterate $x_{k+1}$ satisfies*

$$\frac{1}{2}\|x_{k+1} - x^\star\|_2^2 - \frac{1}{2M}\sum_{i=1}^{n}M_i\|z_m^i - x^\star\|_2^2 \leq \frac{1}{M}(R(x^\star) - R(x_{k+1})) + \frac{1}{M}f(x^\star) + \frac{\delta}{M}$$

$$- \frac{1}{M}\sum_{i=1}^{n}(F_i(z_m^i;\xi_m^i) + \langle G_i(z_m^i;\xi_m^i), x_{k+1} - z_m^i\rangle + \frac{M_i}{2}\|x_{k+1} - z_m^i\|_2^2)$$

$$+ \frac{1}{M}\sum_{i=1}^{n}\sum_{j=1}^{m}\bar{\lambda}_{ij}(F_i(x^\star;\xi_j^i) - f_i(x^\star)).$$

*Proof.* Note that

$$\sum_{i=1}^{m}\sum_{j=1}^{m}\bar{\lambda}_{ij}\langle G_i(z_j^i;\xi_j^i), x^\star - x_{k+1}\rangle = \sum_{i=1}^{m}\sum_{j=1}^{m}\bar{\lambda}_{ij}(\langle G_i(z_j^i;\xi_j^i), x^\star - z_j^i\rangle + \langle G_i(z_j^i;\xi_j^i), z_j^i - x_{k+1}\rangle)$$

$$\leq \sum_{i=1}^{n}\sum_{j=1}^{m}\bar{\lambda}_{ij}\big(F_i(x^\star;\xi_j^i) - F_i(z_j^i;\xi_j^i) + \langle G_i(z_j^i;\xi_j^i), z_j^i - x_{k+1}\rangle\big)$$

$$= \sum_{i=1}^{n}\sum_{j=1}^{m}\bar{\lambda}_{ij}F_i(x^\star;\xi_j^i) - \sum_{j=1}^{m}\bar{\lambda}_{ij}(F_i(z_j^i;\xi_j^i) + \langle G_i(z_j^i;\xi_j^i), x_{k+1} - z_j^i\rangle)$$

$$\leq \sum_{i=1}^{n}\sum_{j=1}^{m}\bar{\lambda}_{ij}F_i(x^\star;\xi_j^i) - \sum_{i=1}^{n}\check{f}_i(x_{k+1};\boldsymbol{\xi}) + \delta.$$

In the first inequality above we used the star-convexity of the oracle (see Assumption 4.10 for the stochastic case and Assumption 4.4 for the deterministic case). In the second inequality we used Lemma A.2.

Inserting this into Lemma A.3 yields

$$\frac{1}{2}\|x_{k+1} - x^\star\|_2^2 - \frac{1}{2M}\sum_{i=1}^{n}M_i\|z_m^i - x^\star\|_2^2 \leq \frac{1}{M}(R(x^\star) - R(x_{k+1})) + \frac{1}{M}\sum_{i=1}^{n}\sum_{j=1}^{m}\bar{\lambda}_{ij}F_i(x^\star;\xi_j^i)$$

$$- \frac{1}{M}\sum_{i=1}^{n}\big(\check{f}_i(x_{k+1};\boldsymbol{\xi}) + \frac{M_i}{2}\|x_{k+1} - z_m^i\|_2^2\big) + \frac{\delta}{M}.$$

The result now follows from adding and subtracting $(1/L)f(x^\star)$ from the right side of the inequality and dropping all but the most recent cut for every piecewise linear model. $\square$

We now distinguish between the deterministic and stochastic case. The following result from Feyzmahdavian et al. (2014) will be useful.

**Lemma A.5.** *Let $(V_k)_{k=0}^{\infty}$ be a non-negative sequence satisfying*

$$V_{k+1} \leq qV_k + p \max_{(k-\tau)_+ \leq \ell \leq k} V_\ell + r, \qquad k = 0, 1, 2, \ldots$$

*for some non-negative constants $p$, $q$ and $r$. If $q + p < 1$, then*

$$V_k \leq \rho^k V_0 + \epsilon, \qquad k = 0, 1, 2, \ldots,$$

*where $\rho = (p + q)^{1/(1+\tau)}$, $\epsilon = r/(1 - p - q)$ and $(k - \tau)_+ = \max\{k - \tau, 0\}$.*

Lemma A.5 will be used to analyze the algorithm under the quadratic functional growth assumption. For the analysis of the convex case, we present a new sequence result that may be of independent interest.

**Lemma A.6.** *Suppose that $(V_k)_{k=0}^{\infty}$ and $(W_k)_{k=0}^{\infty}$ are non-negative sequences satisfying*

$$V_{k+1} \leq \max_{(k-\tau)_+ \leq \ell \leq k} V_\ell - W_{k+1} + r, \qquad k = 0, 1, 2, \ldots \tag{16}$$

*for a non-negative constant $r$. Then, for any $k \geq 1$,*

$$\min_{t \leq k} W_t \leq \frac{(\tau + 1)V_0}{k} + r. \tag{17}$$

*Proof.* We prove (17) by contradiction. Suppose that for some $K \geq 1$, (17) fails to hold. Then, for all $k \leq K$,

$$W_k > \frac{(\tau + 1)V_0}{K} + r. \tag{18}$$

Define $\mathcal{I}_0 = \{0\}$ and for any $t \geq 1$,

$$\mathcal{I}_t = [(t-1)(\tau+1) + 1, t(\tau+1)].$$

Substituting (18) into (16) gives that for all $k \leq K - 1$,

$$V_{k+1} < \max_{(k-\tau)_+ \leq \ell \leq k} V_\ell - \frac{(\tau+1)V_0}{K}. \tag{19}$$

Let $\tilde{t} = \lfloor K/(\tau+1) \rfloor$. For any $t \leq \tilde{t}$, $\mathcal{I}_t \subseteq [0, K]$. Then, using (19), we can derive that for all $t \leq \tilde{t} - 1$,

$$\max_{k \in \mathcal{I}_{t+1}} V_k < \max_{k \in \mathcal{I}_t} V_k - \frac{(\tau+1)V_0}{K}.$$

Summing the above equation over $t \in [0, \tilde{t} - 1]$ and noting that $\max_{k \in \mathcal{I}_0} V_k = V_0$ yields

$$\max_{k \in \mathcal{I}_{\tilde{t}}} V_k < V_0 - \frac{\tilde{t}(\tau+1)V_0}{K}. \tag{20}$$

Note that $\tilde{t} = \lfloor K/(\tau+1) \rfloor$. If $\tilde{t} = K/(\tau+1)$, then

$$\frac{\tilde{t}(\tau+1)V_0}{K} = V_0,$$

substituting which into (20) yields $\max_{k \in \mathcal{I}_{\tilde{t}}} V_k < 0$, which cannot be true because $V_k \geq 0$ for all $k \geq 0$.

If $\tilde{t} < K/(\tau+1)$, then we have $K > \tilde{t}(\tau+1)$. Then, by (19),

$$V_K < \max_{k \in \mathcal{I}_{\tilde{t}}} V_k - \frac{(\tau+1)V_0}{K}.$$

Moreover, since $\tilde{t} \geq K/(\tau+1) - 1$, from (20) we have

$$\max_{k \in \mathcal{I}_{\tilde{t}}} V_k < \frac{(\tau+1)V_0}{K}.$$

Combining the above two equations, we obtain

$$V_K < 0,$$

which cannot hold.

Concluding the above, (17) holds for all $k \geq 1$. $\qquad\square$

EXACT FUNCTION VALUES AND GRADIENTS

First we prove convergence for exact (full-batch) function values and gradients.

**Theorem A.7.** *Under Assumptions 4.1, 4.2, 4.3, and 4.4 the iterates of Algorithm 1 using $M_i = L_i$, $i = 1, \ldots, n$ satisfy*

$$\|x_k - x^\star\|_2^2 \leq \rho^k \|x_0 - x^\star\|_2^2 + \epsilon_\delta, \tag{21}$$

*where $\rho = (L/(L+\mu))^{1/(1+\tau)}$ and $\epsilon_\delta = 2\delta/\mu$.*

*Proof.* We insert $M_i = L_i$, $M = L$ and $F_i(x^\star; \xi_j^i) = f_i(x^\star)$ into Lemma A.4 and use that $f_i$ is smooth with parameter $L_i$ (see Assumption 4.2) to conclude that

$$
\begin{aligned}
\frac{1}{2}\|x_{k+1} - x^\star\|_2^2 - \frac{1}{2L}\sum_{i=1}^n L_i \|z_m^i - x^\star\|_2^2 &\leq \frac{1}{L}(R(x^\star) - R(x_{k+1})) + \frac{1}{L}f(x^\star) + \frac{\delta}{L} \\
&\quad - \frac{1}{L}\sum_{i=1}^n (f_i(z_m^i) + \langle \nabla f_i(z_m^i), x_{k+1} - z_m^i \rangle + \frac{L_i}{2}\|x_{k+1} - z_m^i\|_2^2) \\
&\leq \frac{1}{L}(R(x^\star) - R(x_{k+1})) + \frac{1}{L}f(x^\star) + \frac{\delta}{L} - \frac{1}{L}f(x_{k+1}) \\
&= \frac{1}{L}(F(x^\star) - F(x_{k+1})) + \frac{\delta}{L}.
\end{aligned}
$$

Now suppose that the quadratic functional growth condition of $F$ (see Assumption 4.3) holds. We conclude that

$$\frac{1}{2}\|x_{k+1} - x^\star\|_2^2 - \frac{1}{2L}\sum_{i=1}^n L_i\|z_m^i - x^\star\|_2^2 \leq -\frac{\mu}{2L}\|x_{k+1} - x^\star\|_2^2 + \frac{\delta}{L}.$$

After rearranging terms we get

$$\|x_{k+1} - x^\star\|_2^2 \leq \frac{1}{L+\mu}\sum_{i=1}^n L_i\|z_m^i - x^\star\|_2^2 + \frac{2\delta}{L+\mu}$$

$$\leq \frac{L}{L+\mu} \cdot \max_{(k-\tau)_+ \leq \ell \leq k}\|x_\ell - x^\star\|_2^2 + \frac{2\delta}{L+\mu}.$$

Applying Lemma A.5 yields the desired result. $\qquad\qquad\square$

We now consider the case without the growth assumption.

**Theorem A.8.** *Under Assumptions 4.1, 4.2, and 4.4, the iterates of Algorithm 1 using $M_i = L_i$, $i = 1, \ldots, n$, satisfy that for any $k \geq 1$,*

$$\min_{t \leq k} F(x_t) - F(x^\star) \leq \frac{(\tau+1)L\|x_0 - x^\star\|_2^2}{2k} + \delta. \tag{22}$$

*Proof.* From the proof of Theorem A.7 we know that

$$\frac{1}{2}\|x_{k+1} - x^\star\|_2^2 - \frac{1}{2L}\sum_{i=1}^n L_i\|z_m^i - x^\star\|_2^2 \leq \frac{1}{L}(F(x^\star) - F(x_{k+1})) + \frac{\delta}{L}.$$

Note that $\|z_m^i - x^\star\|^2 \leq \max_{(k-\tau)_+ \leq \ell \leq k}\|x_\ell - x^\star\|^2$ for any $i = 1, \ldots, n$. Equation (22) follows by applying Lemma A.6 with $V_\ell = \frac{1}{2}\|x_\ell - x^\star\|^2$, $W_{k+1} = \frac{1}{L}(F(x_{k+1}) - F(x^\star))$ and $r = \frac{\delta}{L}$. $\quad\square$

STOCHASTIC FUNCTION VALUES AND GRADIENTS

Next we prove the convergence for stochastic function values and gradients.

**Theorem A.9.** *Consider Algorithm 1 with $M_i = \alpha L_i$, $i = 1, \ldots, n$ where $\alpha > 1$. Assume that stochastic function values and gradients are used. Under Assumptions 4.1, 4.2, and 4.10, the iterates of Algorithm 1 satisfy*

$$\min_{t \leq k} \mathbf{E}[F(x_t)] - F(x^\star) \leq \frac{\alpha(\tau+1)L\|x - x_0\|_2^2}{2k} + \epsilon, \tag{23}$$

*where $\epsilon = \epsilon_\delta + \epsilon_{\sigma_1} + \epsilon_{\sigma_2}$ with*

$$\epsilon_\delta = \delta, \qquad \epsilon_{\sigma_1} = n\sigma_1\sqrt{m}, \qquad \epsilon_{\sigma_2} = \frac{\sigma_2^2}{2(\alpha-1)} \cdot \sum_{i=1}^n \frac{1}{L_i}.$$

*If, in addition, Assumption 4.3 holds, then*

$$\mathbf{E}[\|x_k - x^\star\|_2^2] \leq \rho^k\|x_0 - x^\star\|_2^2 + 2\epsilon/\mu,$$

*where $\rho = (\alpha L/(\alpha L + \mu))^{1/(1+\tau)}$.*

*Proof.* According to Lemma A.4 we have

$$\frac{1}{2}\|x_{k+1} - x^\star\|_2^2 - \frac{1}{2M}\sum_{i=1}^n M_i\|z_m^i - x^\star\|_2^2 \leq \frac{1}{M}(R(x^\star) - R(x_{k+1})) + \frac{1}{M}f(x^\star) + \frac{\delta}{M}$$

$$-\frac{1}{M}\underbrace{\sum_{i=1}^n F_i(z_m^i; \xi_m^i) + \langle G_i(z_m^i; \xi_m^i), x_{k+1} - z_m^i\rangle + \frac{M_i}{2}\|x_{k+1} - z_m^i\|_2^2}_{\triangleq T_1}$$

$$+\frac{1}{M}\sum_{i=1}^n\sum_{j=1}^m \bar{\lambda}_{ij}(F_i(x^\star; \xi_j^i) - f_i(x^\star)). \tag{24}$$

Using the Cauchy-Schwarz inequality together with $M_i = \alpha L_i$ and the assumption that worker $i$ is smooth with parameter $L_i$, we bound $T_1$ according to (this holds for all $i = 1, \ldots, n$)

$$
\begin{aligned}
T_1 &= f_i(z_m^i) + \langle \nabla f_i(z_m^i), x_{k+1} - z_m^i \rangle + \frac{L_i}{2} \|x_{k+1} - z_m^i\|_2^2 \\
&\quad + F_i(z_m^i; \xi_m^i) - f_i(z_m^i) + \langle G_i(z_m^i; \xi_m^i) - \nabla f_i(z_m^i), x_{k+1} - z_m^i \rangle + \frac{(\alpha - 1)L_i}{2} \|x_{k+1} - z_m^i\|_2^2 \\
&\geq f_i(x_{k+1}) + F_i(z_m^i; \xi_m^i) - f_i(z_m^i) + \frac{(\alpha - 1)L_i}{2} \|x_{k+1} - z_m^i\|_2^2 \\
&\quad - \|G_i(z_m^i; \xi_m^i) - \nabla f_i(z_m^i)\|_2 \|x_{k+1} - z_m^i\|_2 \\
&\geq f_i(x_{k+1}) + F_i(z_m^i; \xi_m^i) - f_i(z_m^i) - \frac{1}{2(\alpha - 1)L_i} \|G_i(z_m^i; \xi_m^i) - \nabla f_i(z_m^i)\|_2^2,
\end{aligned}
$$

where we in the last inequality used that $(b/2)t^2 - at \geq -a^2/(2b)$ for all $t \in \mathbf{R}$, $a \in \mathbf{R}$ and $b > 0$. Inserting this bound into (24) allows us to conclude that

$$
\begin{aligned}
&\frac{1}{2} \|x_{k+1} - x^\star\|_2^2 - \frac{1}{2M} \sum_{i=1}^n M_i \|z_m^i - x^\star\|_2^2 \leq \frac{1}{M}(R(x^\star) - R(x_{k+1})) + \frac{1}{M} f(x^\star) + \frac{\delta}{M} \\
&\quad - \frac{1}{M} \sum_{i=1}^n \left( f_i(x_{k+1}) + F_i(z_m^i; \xi_m^i) - f_i(z_m^i) - \frac{1}{2(\alpha - 1)L_i} \|G_i(z_m^i; \xi_m^i) - \nabla f_i(z_m^i)\|_2^2 \right) \\
&\quad + \frac{1}{M} \sum_{i=1}^n \sum_{j=1}^m \bar{\lambda}_{ij}(F_i(x^\star; \xi_j^i) - f_i(x^\star)).
\end{aligned}
$$

By taking expectations conditioned on all randomness up to the current iteration and using that $\mathbf{E}[F_i(z_m^i; \xi_i^m)] = f_i(z_m^i)$ where the expectation is conditional, we get

$$
\begin{aligned}
&\frac{1}{2} \mathbf{E}[\|x_{k+1} - x^\star\|_2^2] - \frac{1}{2M} \sum_{i=1}^n M_i \|z_m^i - x^\star\|_2^2 \leq \frac{1}{M} \mathbf{E}[F(x^\star) - F(x_{k+1})] + \frac{\delta}{M} \\
&\quad + \frac{1}{M} \sum_{i=1}^n \frac{1}{2(\alpha - 1)L_i} \underbrace{\mathbf{E}[\|G_i(z_m^i; \xi_m^i) - \nabla f_i(z_m^i)\|_2^2]}_{\triangleq T_2} + \frac{1}{M} \sum_{i=1}^n \sum_{j=1}^m \bar{\lambda}_{ij}(F_i(x^\star; \xi_j^i) - f_i(x^\star)).
\end{aligned}
$$

From the assumption of bounded variance (see Assumption 4.10) we have $T_2 \leq \sigma_2^2$. Hence,

$$
\begin{aligned}
&\frac{1}{2} \mathbf{E}[\|x_{k+1} - x^\star\|_2^2] - \frac{1}{2M} \sum_{i=1}^n M_i \|z_m^i - x^\star\|_2^2 \leq \frac{1}{M} \mathbf{E}[F(x^\star) - F(x_{k+1})] + \frac{\delta}{M} \\
&\quad + \frac{\sigma_2^2}{2(\alpha - 1)M} \sum_{i=1}^n \frac{1}{L_i} + \frac{1}{M} \sum_{i=1}^n \sum_{j=1}^m \bar{\lambda}_{ij}(F_i(x^\star; \xi_j^i) - f_i(x^\star)).
\end{aligned}
$$

After taking expectations again and using the tower property of conditional expectations we get

$$
\begin{aligned}
&\frac{1}{2} \mathbf{E}[\|x_{k+1} - x^\star\|_2^2] - \frac{1}{2M} \sum_{i=1}^n M_i \mathbf{E}[\|z_m^i - x^\star\|_2^2] \leq \frac{1}{M} \mathbf{E}[F(x^\star) - F(x_{k+1})] + \frac{\delta}{M} \\
&\quad + \frac{\sigma_2^2}{2(\alpha - 1)M} \sum_{i=1}^n \frac{1}{L_i} + \frac{1}{M} \sum_{i=1}^n \underbrace{\mathbf{E}\left[ \sum_{j=1}^m \bar{\lambda}_{ij}(F_i(x^\star; \xi_j^i) - f_i(x^\star)) \right]}_{\triangleq T_3}.
\end{aligned}
$$

To bound $T_3$ we note that for $i = 1, \ldots, n$, by the Cauchy-Schwarz inequality, it holds that

$$
\begin{aligned}
T_3 &\leq \mathbf{E}\left[\sum_{j=1}^{m} \bar{\lambda}_{ij} \cdot |F_i(x^\star, \xi_j^i) - f_i(x^\star)|\right] \\
&\leq \left(\mathbf{E}\left[\sum_{j=1}^{m} \bar{\lambda}_{ij}^2\right]\right)^{1/2} \cdot \left(\mathbf{E}\left[\sum_{j=1}^{m} (F_i(x^\star; \xi_j^i) - f_i(x^\star))^2\right]\right)^{1/2} \\
&\leq \left(\sum_{j=1}^{m} \mathbf{E}[(F_i(x^\star; \xi_j^i) - f_i(x^\star))^2]\right)^{1/2} \\
&\leq \sigma_1 \sqrt{m}.
\end{aligned}
$$

Hence,

$$
\frac{1}{2} \mathbf{E}[\|x_{k+1} - x^\star\|_2^2] - \frac{1}{2M} \sum_{i=1}^{n} M_i \mathbf{E}[\|z_m^i - x^\star\|_2^2] \leq \frac{1}{M} \mathbf{E}[F(x^\star) - F(x_{k+1})] + \frac{\delta}{M}
$$

$$
+ \frac{\sigma_2^2}{2(\alpha - 1)M} \sum_{i=1}^{n} \frac{1}{L_i} + \frac{n\sigma_1 \sqrt{m}}{M}.
$$

Note that $\mathbf{E}[\|z_m^i - x^\star\|^2] \leq \max_{(k-\tau)_+ \leq \ell \leq k} \mathbf{E}[\|x_\ell - x^\star\|^2]$ for any $i = 1, \ldots, n$. Then, by Lemma A.6 with $V_\ell = \frac{1}{2} \mathbf{E}[\|x_\ell - x^\star\|^2]$, $W_{k+1} = \frac{1}{M}(\mathbf{E}[F(x_{k+1}) - F(x^\star)])$ and $r = \frac{\delta}{M} + \frac{\sigma_2^2}{2(\alpha-1)M} \sum_{i=1}^{n} \frac{1}{L_i} + \frac{n\sigma_1\sqrt{m}}{M}$, we have (23)

Now further assume that the quadratic functional growth of $F$ (see Assumption 4.3) holds. We have

$$
\frac{1}{2} \mathbf{E}[\|x_{k+1} - x^\star\|_2^2] - \frac{1}{2M} \sum_{i=1}^{n} M_i \mathbf{E}[\|z_m^i - x^\star\|_2^2] \leq -\frac{\mu}{2M} \mathbf{E}[\|x_{k+1} - x^\star\|_2^2] + \frac{\delta}{M}
$$

$$
+ \frac{\sigma_2^2}{2(\alpha - 1)M} \sum_{i=1}^{n} \frac{1}{L_i} + \frac{n\sigma_1 \sqrt{m}}{M}.
$$

Rearranging the terms shows that

$$
\begin{aligned}
\mathbf{E}[\|x_{k+1} - x^\star\|_2^2] &\leq \frac{1}{M + \mu} \sum_{i=1}^{n} M_i \mathbf{E}[\|z_m^i - x^\star\|_2^2] + \frac{2}{M + \mu}\left(\delta + n\sigma_1\sqrt{m} + \frac{\sigma_2^2}{2(\alpha - 1)} \sum_{i=1}^{n} \frac{1}{L_i}\right) \\
&\leq \frac{M}{M + \mu} \cdot \max_{(k-\tau)_+ \leq \ell \leq k} \mathbf{E}[\|x_\ell - x^\star\|_2^2] + \frac{2}{M + \mu}\left(\delta + n\sigma_1\sqrt{m} + \frac{\sigma_2^2}{2(\alpha - 1)} \sum_{i=1}^{n} \frac{1}{L_i}\right).
\end{aligned}
$$

Applying Lemma A.5 yields the desired result. □

## A.3 SOLVING THE MASTER PROBLEM

Here we discuss the complexity of solving the master problem (3) approximately. We use an accelerated projected gradient method to solve the dual problem

$$
\begin{aligned}
\text{minimize} \quad & g(\lambda) \\
\text{subject to} \quad & \lambda \in \Delta.
\end{aligned} \tag{25}
$$

The objective function $g(\lambda)$ is defined in Lemma 3.1 and $\Delta \subseteq \mathbf{R}^{mn}$ is the Cartesian product of $n$ probability simplices of dimension $m$. In each iteration we must project onto $\Delta$, which can be done at a cost of order $\mathcal{O}(nm \log m)$. The cost for evaluating the gradient (10) of the objective function $g(\lambda)$ is dominated by a term of order $\mathcal{O}(nmd)$, in addition to the cost of evaluating the proximal operator of $R$. (We recall that $n$ is the number of workers, $m$ is the bundle size, and $d$ is the dimension of $x$.) If, for example, $R(x) = \lambda \|x\|_1$, then the cost of evaluating the proximal operator is $\mathcal{O}(d)$, so in this case the total cost per iteration of the projected gradient method is dominated by a term of order $\mathcal{O}(md)$. In practice we found that only a dozen of iterations was often sufficient to satisfy the termination criteria (6).

Table 2: Properties of the datasets that we use. The total number of data points is denoted by $N$ and $d$ is the dimension of the decision variable. The column labeled DENSITY shows the percentage of non-zero entries. The label DENSE means that the data matrix is stored as a dense matrix.

| DATASET | | $N$ | $d$ | DENSITY | $\lambda_1$ |
|---|---|---|---|---|---|
| MNIST8M | (BINARY) | 164 8890 | 784 | DENSE | $3e$-$3$ |
| RCV1 | (BINARY) | 677 399 | 47236 | 0.15% | $3e$-$6$ |
| EPSILON | (BINARY) | 500 000 | 2000 | DENSE | $5e$-$5$ |
| SVHN | (MULTICLASS) | 630 420 | 10240 | DENSE | $1e$-$3$ |
| MNIST8M | (MULTICLASS) | 8100 000 | 7840 | DENSE | $8e$-$3$ |

Modern interior-point solvers are fast, easy-to-use, robust, and good at exploiting sparsity. It is therefore natural to use an interior-point solver for solving the master problem (3). If $R(x) = \lambda\|x\|_1$, the master problem can be formulated as a quadratic program with a separable objective function and a coefficient matrix that is quite sparse (see, for example, Andersen et al. (2011)).

To investigate the impact of our specialized approach for solving the subproblem, we ran ABM twice and solved the master problem with either the projected gradient method applied to the dual (25) using accuracy $\delta = 10^{-7}$, or the state-of-the-art interior-point solver Clarabel (Goulart & Chen, 2021) applied to the quadratic programming formulation of (3). For mnist8m, rcv1 and epsilon, the average time to solve the master problem was 0.017, 0.29, and 0.027 seconds for the gradient method, versus 0.20, 13.4, and 0.66 seconds for Clarabel. In other words, the gradient method (implemented in Python) is more than an order of magnitude faster. This comparison is not completely fair, since Clarabel in general finds a solution with higher accuracy. However, as shown in Theorem 4.5, it is not necessary to solve the master problem exactly to maintain convergence guarantees.

### A.4 ADDITIONAL NUMERICAL EXPERIMENTS

#### DIMENSION OF DATA SETS

Table 2 shows the dimensions of the data sets and the value of the regularization parameter $\lambda_1$.

#### CONVEX EXPERIMENTS

Figure 3 shows the performance of ABM, DAve-RPG and PIAG for binary logistic regression with $\lambda_2 = 0$, *i.e.,* when the objective function is convex but not strongly convex. The optimization trajectories are very similar to the progress in the strongly convex case for mnist8m and epsilon, but removing the strong convexity degrades the performance on rcv1 for all three methods (cf. Figure 1 in the main text). Nevertheless, ABM outperforms the two competitors.

#### BENCHMARKING ON MULTINOMIAL LOGISTIC REGRESSION

For the multiclass classification problems the objective function is

$$f(x) = -\frac{1}{N} \sum_{j=1}^{N} \sum_{k=1}^{K} \mathbf{1}\{y_j = k\} \log\left(\frac{e^{x_k^T a_j}}{\sum_{\ell=1}^{K} \exp(x_\ell^T a_j)}\right) + \frac{\lambda_2}{2} \sum_{k=1}^{K} \|x_k\|_2^2$$

$$R(x) = \lambda_1 \sum_{k=1}^{K} \|x_k\|_1,$$

where $y_1, \ldots, y_N \in \{1, 2, \ldots, K\}$ are the labels. Here the decision variable is $x = (x_1, \ldots, x_K)$ where each $x_j$, $1 \leq j \leq K$ is a vector with dimension equal to the number of features. Figure 4 shows the relative suboptimality of ABM, DAve-RPG and PIAG on the two multiclass datasets SVHN and mnist8m. (The dimension of the problems can be found in Table 2.)

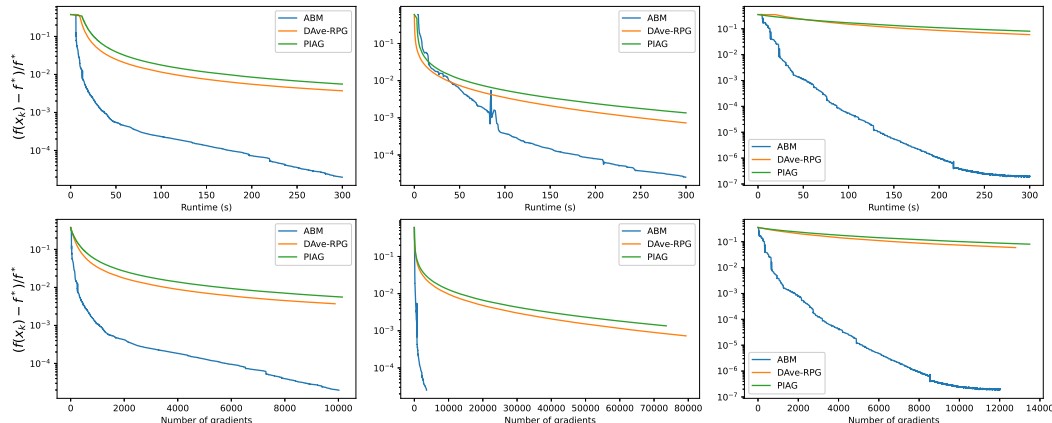

Figure 3: The progress of `ABM`, `DAve-RPG` and `PIAG` on the binary classification problems for $\lambda_2 = 0$. The datasets are arranged in the order `mnist8m`, `rcv1` and `epsilon` from the left.

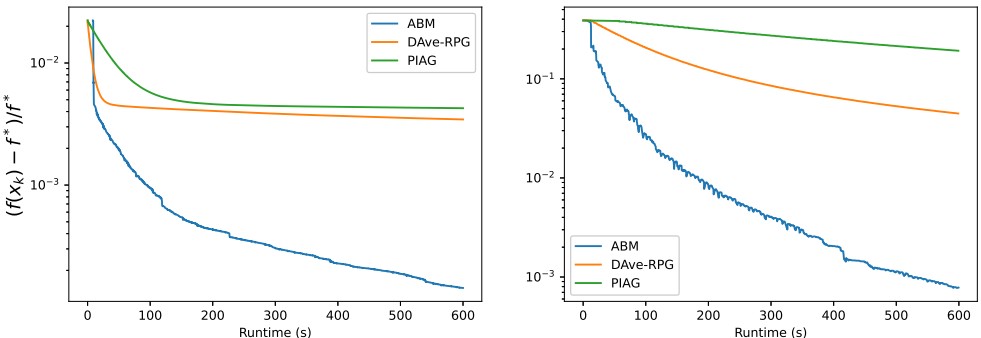

Figure 4: The progress of ABM, DAve-RPG and PIAG on the multiclass classification problems. The datasets are arranged in the order `SVHN` and `mnist8m` from the left.

STOCHASTIC FUNCTION VALUES AND SUBGRADIENTS

We test the stochastic extension of `ABM` using mini-batches for multiclass classification on the data set `mnist8m`. We split each worker's data set into 100 mini-batches. For `ABM` based on exact function and gradient evaluations we estimate the smoothness parameters as described in §5. For the stochastic variant using mini-batches we estimate the smoothness parameters as follows. Each worker stores the last point, say $z_{m-1}^i$, it was queried in, and when queried again in a point $z_m^i$, the worker draws $\xi$ representing a mini-batch and then evaluates both $G_i(z_m^i; \xi)$ and $G_i(z_{m-1}^i; \xi)$. The worker can then estimate the smoothness parameter $L_i$ with

$$\hat{L}_i = \frac{\|G_i(z_m^i; \xi) - G_i(z_{m-1}^i; \xi)\|_2}{\|z_m^i - z_{m-1}^i\|_2}.$$

The worker then sends back both the gradient $G_i(z_m^i; \xi)$ and the smoothness estimate $\hat{L}_i$ to the central server.

For comparison we also implemented a synchronous proximal stochastic gradient method. The step size parameter was carefully tuned. The left part of Figure 5 shows the progress of `ABM`, the stochastic extension (`ABMStoch`), and the proximal stochastic gradient method (`ProxSGD`). The right part of Figure 5 shows the progress in the presence of some struggling workers. (For the right part we let one third of the workers have random delays uniformly distributed in the interval $[2t_{\text{grad}}, 4t_{\text{grad}}]$ every time they compute a gradient, where $t_{\text{grad}}$ is the time required to compute the gradient.) It is interesting to note that while `ProxSGD` performs much worse for simulated delays, both `ABM` and `ABMStoch` are barely affected by the delays.

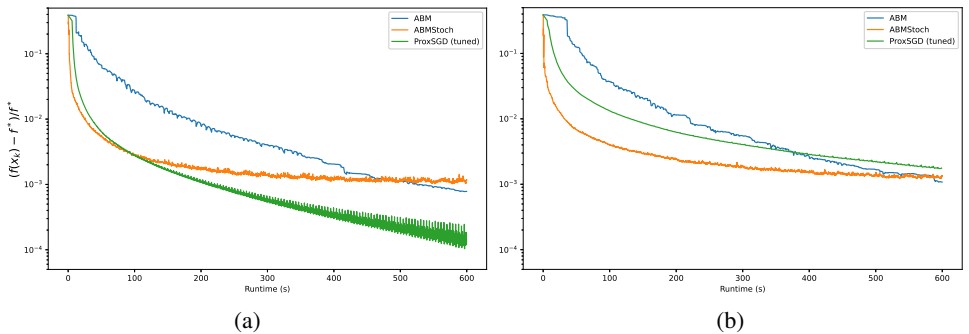

(a)                    (b)

Figure 5: **Left**: The progress of `ABM`, the stochastic variant `ABMStoch` and a tuned synchronous proximal stochastic gradient method `ProxSGD`. **Right**: The progress under simulated delays.

