# OpenReview forum: "An Asynchronous Bundle Method for Distributed Learning Problems"
_ICLR.cc/2025/Conference — ICLR 2025 Poster_

### Official Review · Reviewer_4dTz · 2024-10-19

**Soundness:** 4
**Presentation:** 4
**Contribution:** 3
**Rating:** 8
**Confidence:** 5

**Summary:**

The paper presents a bundle method for asynchronous distributed optimization. By increasing the server-side complexity, it boosts the performance. The algorithm operates under unknown (bounded) delays, while favorable is the effort for automated parameter tuning.

**Strengths:**

+ Useful algorithmic contributions (especially embodying parameter auto-tuning).
+ Rigorous analysis (incorporating cases of practical interest such as inexact server solutions and mini-batch processing).
+ Well-written and clear paper (especially commendable is that the authors provide insights for all choices as well as their analysis).
+ Rich, well-presented, and convincing experiments.

**Weaknesses:**

- Dubious scalability in large systems ($n \gg 1$).

Others--outweighed by the strengths--are that the contribution is an extension of previous works and that the code is not presently shared.

**Questions:**

Questions:

1. Your analysis in (7) applies across a subsequence ($\exists k\ge K$). Can this be strengthened? For example, can you make assertions for the convergence of $\min_{j\le k} \{F(x_j) - F(x^\star)\}$? How does it compare with the literature?
Can you also comment on the reason for the discrepancy between (7) and (8) in terms of the same (the latter applies $\forall k$).
2. Can you include the adaptive tuning aspect ($\hat{L}_i$) in the analysis, or provide a remark sketching the same?
3. Can you please comment on means to boost scalability in large systems ($n \gg 1$)?


Suggestions:

- Please include the cost of solving the master problem ($O(nmd)$) in the main paper.
- Can you rephrase the statement in lines 362-365? It seems a bit sloppy for your fluent presentation style.
- Please explain the reasoning in line 458 (why in this case the master problem is more challenging).
- Please elaborate on "potentially discarding outdated information" in line 532 (in relation to dynamic bundle size selection).

Editing:

- line 55: "insensitive" does not seem accurate and can be misleading; please rephrase.
- line 88: counterpart -> counterparts.
- lines 108-111: please rephrase to prevent any misperception of a contradictory statement (gradients provide local; in contrast, gradients provide global).
- I suggest numbering the equation in line 207.
- line 256: the central server for each worker $i$ -> each worker $i$ (to prevent any confusion that the server has access to user data).
- line 295: delete "weak".
- line 314: rephrase "gracefully".
- line 490: no tuning -> no tuning in these experiments (to avoid overstating).

---

> ### Author Response · Authors · 2024-11-23
>
> We are very grateful for your detailed feedback. Thank you! Below, we provide detailed responses to each of your questions.
>
> **Analysis.** Thank you for asking about more details on the analysis.
> * **Can we strengthen the analysis in the absence of the growth condition?** We are happy to announce that we have significantly improved the convergence result in the absence of the growth condition (see Theorem 4.9 of the modified manuscript). We now prove sublinear convergence to a neighborhood of the optimal solution, rather than the convergence of a subsequence.
> * **What causes the discrepancy between the results with and without the growth condition?** In both cases, with and without the quadratic functional growth condition, we can establish the inequality
> $$
> ||x_{k+1} - x^\star ||^2 \leq (2/L)  (F(x^\star) - F(x_{k+1})) + \max_{(k - \tau)_+ \leq l \leq k} || x_l - x^\star ||^2 + \delta/L
> $$
>
>     When the growth condition holds, we can directly relate the function value gap $F(x^\star) - F(x_{k+1})$ to the iterate gap $||x_{k+1} - x^\star ||^2$. This connection enables us to derive a contraction property, ultimately proving linear convergence. In the absence of the growth condition, however, this direct relationship no longer holds. Instead, we must rely on an alternative approach, proving and applying a novel sequence result (Lemma 4.8 in the modified manuscript). Sequence results are generally weaker and more flexible tools, which leads to a looser analysis.
> * **How do our results compare to the literature?** Under the growth condition, our asynchronous bundle method is the only one with provable linear convergence to a neighborhood of the optimal solution. Without the growth condition, our asynchronous bundle method is the only existing one with a sublinear convergence rate. (Existing asynchronous bundle methods only gurantee convergence of a subsequence. For example, the work [1] shows in their Theorem 1 that their asynchronous bundle method satisfies $\min_{j \leq k} F(x_j) \to F(x^\star)$, assuming that their subproblem is solved exactly. We should point out that unlike us, [1] assumes the existence of a convex compact constraint set and also that the local loss functions of the workers are Lipschitz. In comparison, we assume that the local loss functions are smooth.)
>
> [1] F. Iutzeler, J. Malick, and W. Oliveira. Asynchronous level bundle methods, 2020.
>
> **Adaptive Tuning Aspect** We appreciate the suggestion to incorporate adaptive smoothness parameter estimation into the analysis and agree that it is an important direction for further research. However, after revisiting our current analysis, we realized that including this aspect would require substantial additional analysis, which goes beyond the scope of the current work. Nevertheless, we will certainly keep this suggestion in mind for future work.
>
>
> **Scalability**. We have commented on scalability in the official comment section above.
>
> **Other comments**. We deeply appreciate the reviewer's meticulous reading of our paper and his or her thoughtful feedback. We have addressed the recommended suggestions and editing tips, which have enhanced the overall quality of the manuscript. Thank you once again for reviewing our paper. Is there anything else we can clarify?

---

> > ### Comment · Reviewer_4dTz · 2024-11-25
> > **Post-rebuttal**
> >
> > The authors have addressed many comments and have improved the manuscript. I maintain my positive evaluation.
> >
> > PS By the way, my question about the discrepancy was about the need for subsequence (the utility of growth condition is standard). Regarding the adaptive tuning aspect and other limitations pointed out by the reviewers, you may want to update the discussion section acccordingly.

---

### Official Review · Reviewer_KLUM · 2024-10-29

**Soundness:** 2
**Presentation:** 3
**Contribution:** 2
**Rating:** 6
**Confidence:** 3

**Summary:**

The manuscript describes a asynchronous bundle method for distributed optimization. The results are presented well and the manuscript is well-written and clear. However, I have the following concerns that somehow limit the scope of the results.

1) Novelty

Bundle methods have been around for quite some time and have been also around in the machine learning community, since the 2000s (there is e.g. a well-cited paper by A. Smola). Similarly, the fact that bundle methods can be used for asynchronous parallel optimization has also been discussed several years ago (around the 2010s). Hence, the conceptual novelty of the work is very limited.

On the technical side, early works seem to have focused on non-smooth optimization (which arguably is also the setting where bundle methods are very adequate for), while the current work looks at (essentially) strongly convex composite objectives that have a smooth and non-smooth part. In contrast to earlier results, the manuscript claims sublinear/linear convergence to a neighborhood of the solution (8). However, the current statement of Theorem 4.5/Theorem 4.10 does not prove linear/sublinear convergence of the iterates, but rather for a subsequence of the iterates.
As such, the theorems do still allow for an arbitrarily slow convergence of the iterates (the variable k can be arbitrarily large). The usefulness of deriving a convergence rate for the subsequence is unclear to me, unless this is used as an intermediate step to argue that the entire sequence converges at the given rate (which is not done).


2) Experimental evaluation

The experiments are limited and consist of (regularized) logistic regression with only a very small set of workers (n=9). This does not seem convincing to a machine learning audience for two reasons: i) the community would be interested in deploying the method at scale (meaning several 1000 workers). ii) the community would likely use more elaborate models (NNs, transformers, etc) for classification.
Can the authors argue about scalability of the method with respect to the number of workers? How does the method empirically perform on nonconvex problems?

--

Minor questions, discussion:
- Can the authors comment on the role of the variable "m" in the convergence results Theorem 4.5/Theorem 4.10? As far as I see, the variable doesn't play a role in the results.

- Is there a good way to reduce the memory overhead? As is, the central server needs to store O(mn) gradients, where n denotes the number of workers. This seems really impractical for large scale problem instances.

**Strengths:**

see above

**Weaknesses:**

see above

**Questions:**

see above

---

> ### Author Response · Authors · 2024-11-23
>
> Thank you for excellent feedback and for reviewing our paper. Below, we address your concerns.
>
> **Concern regarding novelty:** Thank you for raising the concern about novelty. While it is true that bundle methods have been studied extensively, our algorithm has several distinguishing features compared to the paper by A. Smola [1] and existing asynchronous bundle methods [2-5]:
> * **Inexact subproblems**. The methods in [1-5] assume that the subproblem is solved exactly. In contrast, our analysis provides a termination criterion that can be used to derive a *practical* "inexact" subproblem solver. (In Appendix A.3, we show that our subproblem solver is more than an order of magnitude faster than an interior-point solver that computes an "exact" solution. This contribution is therefore significant.)
> * **Natural interpretation of stabilization parameter**. In the proximal bundle methods in [2, 4, 5], the choice of the stabilization parameter for the subproblem is based on heuristics. In contrast, our analysis allows us to directly relate the stabilization parameter to the smoothness of the workers. This relationship results in a more natural and straightforward method for selecting an appropriate value for the stabilization parameter.
> * **Asynchronous updates.** The bundle method proposed in [1] is synchronous, whereas both our algorithm and those in [2–5] support asynchronous updates. However, unlike the asynchronous bundle methods in [2–5], our method and its analysis explicitly quantify how the level of asynchrony of the system affects the convergence rate under the growth assumption. This type of quantification does not appear in [2-5].
> * **Stochastic analysis.** In contrast to [1–5], our analysis covers stochastic function and gradient evaluations.
>
> We believe the distinguishing features above show that our approach is novel and advances the state-of-the-art in both the theory and practice of bundle methods for ML.
>
> [1] Bundle Methods for Regularized Risk Minimization, 2010.
>
> [2] Incremental-like bundle methods with application to energy planning, 2009.
>
> [3] Asynchronous level bundle methods, 2020.
>
> [4] Incremental Bundle Methods using Upper Models, 2018.
>
> [5] Bundle method for exploiting additive structure in difficult optimization problems, 2015.
>
> **Clarification on convergence results:** We appreciate your comments and agree that our convergence results were not presented with sufficient clarity in the original submission. To address this, we have reorganized the manuscript to emphasize:
> * **Linear convergence**: Under the growth condition, we establish linear convergence of the iterates to a neighborhood of the optimal solution in Theorem 4.5 of the modified manuscript. *Our asynchronous bundle method is the only existing one with this provable rate under the growth condition*.
> * **Sublinear convergence**:  Without the growth condition, we have derived a new result that shows sublinear convergence to a neighborhood of the optimal solution (see Theorem 4.9 in the modified manuscript). *Our asynchronous bundle method is the only existing one with a provable sublinear convergence rate without the growth condition*.
>
> **Experimental evaluation**: We commented on scalability in the official comment section above.
>
> **Role of the bundle size $m$:** You are correct that the bundle size $m$ does not appear explicitly in the convergence results. When analyzing bundle-like optimization algorithms, employing a piecewise-linear model with $m > 1$ is not advantageous from the perspective of worst-case complexity analysis [6]. However, empirical evidence suggests that increasing the bundle size from $m = 1$ can significantly improve the performance. (We show this in the paper in Figure 2 where increasing the bundle size from $m = 2$ to $m = 5$ makes our method much faster.)
>
> [6] Y. Nesterov. Primal-dual subgradient methods for convex problems, 2009.
>
> **Memory overhead**: We acknowledge that our method requires more memory at the central server compared to SGD-based methods. However, it uses substantially less memory than other distributed asynchronous algorithms that store Hessian approximations. Importantly, in our method the server only needs to store gradients, not training data, which mitigates memory concerns in many practical setups.  For example, consider a training environment with a modest number of computer nodes where each node is treated as a worker (as described in the official comment above). With bundle size $m=5$ and $n=30$ workers, and an ML model with 10 million parameters, the server requires approximately 6 GB of memory to store the gradients in single precision. In many cases this is substantially less than the memory required by the workers who store the actual training data.
>
> **Once again, thank you for excellent feedback. Is there anything we can clarify? We hope that our answers have clarified the many advantages of our algorithm, and that you are willing to consider raising your score.**

---

> > ### Comment · Reviewer_KLUM · 2024-11-26
> >
> > I thank the reviewers for the clarification and for updating the results in the manuscript. I agree that the linear and sublinear rates derived in Thm. 4.5 and Thm. 4.9 provide a significant improvement to the earlier results.
> >
> > However, I do not necessarily agree with the authors that the extensions to inexact subproblems/stochastic gradients provides a substantial novelty from a technical point of view. I decided to raise my score to weak accept, nonetheless.

---

### Official Review · Reviewer_zZ8S · 2024-11-03

**Soundness:** 3
**Presentation:** 3
**Contribution:** 3
**Rating:** 8
**Confidence:** 3

**Summary:**

The authors propose a new asynchronous optimization method for solving distributed convex problems. The proposed method approximates each local function using a piece-linear model constructed from a fixed-window of past updates. The main advantage of the proposed method is that it does not assume a maximum time delay and takes performs regularization at the server. The experiments demonstrate the effectiveness of the bundle method in asynchronous settings and interestingly shows a significant improvement over standard approaches.

**Strengths:**

The authors provide a strong argument for their methodology. It seems intuitive that using past updates can help estimate the local function

**Weaknesses:**

The reviewer wonders if the authors can provide more detail on how the weighting is chosen. It seems that the weighting should be chosen also based on the time-delay

**Questions:**

The reviewer wonders if the authors can comment if this applies to settings where each worker has an individual time-dela, $\tau$.

---

> ### Author Response · Authors · 2024-11-23
>
> We deeply appreciate your feedback and your questions. Below we address them one by one.
>
> **Question 1:** Provide more details on how the weighting is chosen.
>
> *Response:* To construct the bundle center $\bar{z}$ we weight together the most recent query points for each worker according to
> $$
> \bar{z} = \sum_{i=1}^n w_i z^i_m,
> $$
>
> where $z^i_m$ is the most recent query point for which the server has received information from worker $i$. The *weight* $w_i$ is given by $w_i = L_i / L$, where $L_i$ is the smoothness of worker $i$ and $L = \sum_{i=1}^n L_i$ is the total smoothness.
>
> At first glance, it might seem natural to adjust the weight $w_i$ to account for the actual delay experienced by worker $i$. However, our convergence analysis reveals a surprising result: the method converges robustly with this choice of the bundle center, regardless of the actual delays. Our intuition for why this approach works is that, even if $z^i_m$ is an outdated iterate, the information derived from it remains relevant because it provides a valid lower bound on $f_i$. Thus, we do not need to compensate for the actual delays when weighting together to form the bundle center.
>
> **Question 2:** Do the results apply to settings where each worker has an individual maximum time delay?
>
> *Response:* Excellent question. In the literature it is most common to assume a shared maximum information delay for the workers, but let's assume for a moment that the workers have different maximum delays. Denote the maximum delay for worker $i$ by $\tau_i$. Then we can refine our analysis to prove an inequality of the form
> $$
> || x_{k+1} - x^\star ||^2 \leq \sum_{i=1}^n \beta_i \max_{(k - \tau_i)_+ \leq l \leq k} || x_l - x^\star ||^2 + \epsilon,
> $$
>
> where $\beta_i = L_i/(L + \mu)$ and $\epsilon = 2\delta/(L+\mu).$ This should be compared to our current analysis that is based on a common maximum time delay $\tau$ and the inequality
> $$
> || x_{k+1} - x^\star ||^2 \leq \bigg(\sum_{i=1}^n \beta_i \bigg) \max_{(k - \tau)_+ \leq l \leq k} || x_l - x^\star ||_2^2 + \epsilon.
> $$
>
> Intuitively it feels like the first inequality should give a better convergence rate than the second, but we have not yet been able to prove it rigorously.
>
> **We hope that we have now addressed your questions in a clear way.**

---

> > ### Comment · Reviewer_zZ8S · 2024-11-29
> >
> > Thank you. The authors have answered all the questions and I thank the authors for updating the manuscript.

---

### Official Review · Reviewer_NXCg · 2024-11-05

**Soundness:** 3
**Presentation:** 3
**Contribution:** 3
**Rating:** 6
**Confidence:** 3

**Summary:**

The authors in this paper propose an asynchronous bundle method to solve distributed learning problems with parameter server architecture. The algorithm constructs a piecewise linear model to approximate the local loss of each worker and uses this model to compute the next iterate. This method enjoys a more accurate approximation of the objective function, compared to other first-order asynchronous algorithms. The authors prove that their algorithm can converge to a neighborhood of the solution for both deterministic and stochastic settings. In addtion, they show how the convergence results depend on several key parameters, such as upper delay bound $\tau$, approximation error $\delta$ and noise variance $\sigama_1^2, \sigama_2^2$. They also report some numerical simulations to validate the effectiveness of their proposed algorithm.

**Strengths:**

- The proposed algorithm seems novel to the reviewer, and is fast and easy to tune in practice.
- The authors provide convergence guarantee for their algorithm and clearly show that how some key parameters (e.g., $\tau$, $\delta$, $\sigma_1^2$, $\sigma_2^2$) affect the convergence rate and steady error term.
- The paper is clearly structured and seems technically correct.

**Weaknesses:**

- There are no proper comparisons of theoretical results of the baseline algorithms: i.e., to DAve-RPG[1] and PIAG [2].
- The data heterogeneity analysis is missing. It is unclear how the proposed algorithm will perform in the case of strong data heterogeneity  in the numerical experiments.
- The scalability of the proposed algorithm is not well addreesed; note that only a system with 9 workers is considered in the numerical experiments.

[1] Konstantin Mishchenko, Franck Iutzeler, Jer´ome Malick, and Massih-Reza Amini. A Delay-tolerant Proximal-Gradient Algorithm for Distributed Learning. Proceedings of the 35th International Conference on Machine Learning, 2018.

[2] Xuyang Wu, Sindri Magnusson, Hamid Reza Feyzmahdavian, and Mikael Johansson. Delay-Adaptive Step-sizes for Asynchronous Learning. Proceedings of the 39th International Conference on Machine Learning, 2022.

**Questions:**

- The reviewer would like to know how their algorithm improves the complexity compaerd to the baselines DAve-RPG[1] and PIAG [2].
- How can the upper bound of delay $\tau$ be guaranteed by setting bundle size $m$ in their practical experients, or in ther words, how is the value of $\tau$ related to the value of $m$ in practice?
-  How does the proposed algorithm scale in terms of the node number? Can the authors provide some numerical experiments to illustrate this?
- What’s the limitation of the third one in Assumption 4.9?

**Details Of Ethics Concerns:**

N.A.

---

> ### Author Response · Authors · 2024-11-23
>
> Thank you for your detailed feedback and for reviewing our paper. We are very grateful. Below, we address your concerns.
>
> **Data heterogeneity**:  We agree that we could have been more explicit about data heterogeneity in our initial submission since this is a major advantage of our algorithm. While many of the classical federated learning algorithms exhibit bias when clients have diverse data distributions, our algorithm does not suffer from this phenomenon. However, the data distributions influence the smoothness parameters of the local loss functions, and thereby the convergence rate of the algorithm. Let us elaborate.
>
> For the deterministic version of our method, we prove a convergence rate in Theorem 4.4 that depends on the quantity $L = \sum_{i=1}^n L_i$. In a heterogeneous setting where the $L_i$’s vary, this bound can be much stronger than one based on the quantity $n L_{\max}$, where $L_{\max} = \max_{i=1, \dots, n} L_i$. For example, consider a scenario where the data is distributed unevenly with one worker having much larger smoothness parameter than the others. Such a scenario would be problematic for methods with convergence rates depending on $L_{\max}$. However, our method remains robust to this heterogeneity as long as the sum of the smoothness parameters remains constant. Conceptually similar results are well-known for coordinate descent methods, see eg. [1].
>
> That said, we wish to stress that our method and its analysis works without any strong assumptions on the data distribution, which we believe is a significant advantage.
>
> [1] Hong, M., Wang, X., Razaviyayn, M., and Luo. Z. Iteration complexity analysis of block coordinate descent methods, 2017.
>
>
> **Theoretical comparison to baselines:**
>
> *Analysis under strong convexity-like assumption*. Under the bounded delay assumption in our paper, the convergence rate of DAve-RPG is $((L-\mu)/(L + \mu))^{2/(\tau+1)}.$ Our rate is $(L/(L+\mu))^{1/(\tau+1)}.$ If we consider fixed step-size for PIAG and assume $L\ge 2\mu$, then the linear rate of PIAG becomes
> $$e^{-\frac{3\mu}{26L(\tau+1)}} \approx 1-\frac{3\mu}{26L(\tau+1)}.$$
>
> The first two rates are for the same merit function (squared Euclidean distance to the optimal solution), and the third rate is in terms of the function value gap.
>
> These rates are of the same order with respect to $\tau$ and $\mu/(L+\mu)$, but the rate of DAve-RPG is the best. It is natural that our rate is not as sharp as DAve-RPG's rate, since we considered a much more complex algorithmic setting which makes the analysis more challenging and looser. On the other hand, in the numerical experiments, our method outperforms DAve-RPG.
>
> *Analysis under convexity.* For all three methods, convergence can be guaranteed for general convex problems (note that the DAve-RPG paper has a SIAM Optimization extension [2] discussing convergence for convex functions without assuming strong convexity.) The work on PIAG derived $\mathcal{O}(1/k)$ rate on the function value gap $F(x_k)-F(x^\star)$, the work on DAve-RPG derived
> $\min_{t\le k} ||g_t||^2\le O(1/k)$ for $g_t\in \partial F(x^k),$
> and our new convergence result is of the form
> $\min_{t\le k} F(x_t)-F^\star\le \mathcal{O}(1/k).$
> All these rates are similar.
>
> We should point out that, unlike the work on PIAG and DAve-RPG, our analysis covers stochastic gradients.
>
> [2] Mishchenko, K., Iutzeler, F., Malick, J. A distributed flexible delay-tolerant proximal gradient algorithm, 2020.
>
> **Scalability:** We have commented on your concern on scalability in the official comment section above.
>
> **Relationship between bundle size and maximum delay bound:** The bundle size $m$ does not affect the maximum delay bound $\tau$. Note that $\tau$ is a bound on the delay of the \emph{most recent} query point for each worker (see Assumption 4.1 in the paper). For example, if $m = \infty$ (meaning all previous gradient information is retained in the objective function approximation) and each node provides an update at least every 10th iteration, then $\tau = 10$.
>
> **Limitation of third assumption in Assumption 4.9:** The assumption on the function value noise is not restrictive because 1) the bound is specified in expectation, meaning that the inequality does not need to hold uniformly, and 2) the noise is required to be bounded only at optimal solutions, rather than throughout the entire space.
>
> **Thank you for your excellent feedback. Is there anything else we can clarify? We hope that our answers have addressed your concerns, and that you are willing to consider raising your score.**

---

> ### Comment · Reviewer_NXCg · 2024-11-30
>
> The reviewer thank the authors for their efforts in clarifying and updating the results in the manuscript. The author's reply have addressed most of my comments and questions.  I have no further questions and I am thus willing to raise my score to 6.

---

### Author Response · Authors · 2024-11-23
**A comment on convergence and scalability**

We thank the reviewers for their thoughtful feedback and valuable suggestions. We appreciate their careful evaluation of our work and agree with many of the concerns they have raised. We have modified our manuscript and addressed these points in detail below on a reviewer-specific basis. Additionally, we would like to make a general comment on a new convergence result we have derived and also on the scalability of our method.

**Comment on convergence result:** Two reviewers essentially asked for a stronger convergence result for the case where we don't assume the quadratic functional growth assumption. We are happy to announce that we have been able to derive a novel sequence result that allows us strengthen the analysis significantly. We now prove that the function value gap decreases at a rate $\mathcal{O}(1/k)$ (see Lemma 4.8 and Theorem 4.9 in our modified manuscript).

**Comment on scalability:** We thank the reviewers for raising concerns about the scalability of our algorithm. Scalability is indeed a critical aspect of modern optimization techniques, particularly in Federated learning (FL) and other large-scale distributed settings. Below, we address the scalability challenges and clarify the contexts in which our algorithm excels.

 * Federated learning is used in diverse scenarios, from coordination of massive networks of resource-constrained devices (cross-device FL) to collaborative model training across data-rich institutions (cross-silo FL). It is unlikely that there is a single algorithm that is superior in all scenarios. Different contexts face different challenges and require systems optimized for different performance measures: some prioritize training speed and parallelization, while others focus on robustness and ease of deployment. Our algorithm, at least in its present form, caters more to the cross-silo setting, where it allows for fully asynchronous implementations without manual tuning, and often gives faster training than alternatives.
 * In large-scale distributed training environments with a modest number of computer nodes, each node being equipped with multiple GPUs, it is natural to treat each *node* as a worker. In a training environment like this, it is reasonable to have a couple of tens of nodes or workers, rather than in the order of thousands. Our algorithm is particularly well-suited for this distributed setup, since each worker can fully exploit its available hardware, leveraging as much parallelism as possible, to evaluate its gradient. Additionally, the asynchronous nature of our algorithm ensures that the performance of one node remains unaffected by latency or delays in others, enabling robust training.
 * A potential computational bottleneck of our algorithm, compared to other asynchronous methods, is solving the subproblem at the server. However, in many cross-silo applications, the server has significant computational power. For example, we collaborate with a medical equipment manufacturer that uses federated learning to perform training on data drawn from installations at different sites. In their current solution, the server uses a fast multi-core CPU and constitutes only a small fraction of the overall system cost.

   The main cost of solving the subproblem is evaluating the gradient of the dual objective function (cf. equation (11) in the paper), which is done using matrix-vector multiplication. This can be parallelized effectively with a GPU if solving the subproblem, rather than the communication between the server and the workers, is the computational bottleneck. To further reduce the workload at the server, the bundle size $m$ can be adjusted adaptively. Notably, when $m = 1$, the subproblem has a closed-form solution.

   Furthermore, although we may not have time during the rebuttal period to conduct additional experiments to investigate the empirical performance further, we can provide quantitative insights based on our current experiments and the theoretical cost of solving the subproblem, which is $\mathcal{O}(mnd)$ where $d$ is the dimension of the decision variable, $n$ is the number of workers, and $m$ is the bundle size. For example, for the dataset *epsilon*, our Python implementation of the subproblem solver running on a single CPU takes on average 0.027 seconds for $n = 9$ workers (see appendix A.3 in the manuscript). Given the linear scaling in the number of workers, we could scale up to 300 workers and still be able to solve the subproblem in a second.

We acknowledge the reviewers' concerns regarding scalability and recognize the importance of addressing these challenges in large-scale settings. We hope our detailed discussion above clarifies the contexts where our algorithm excels and sheds light on its potential for scaling effectively.

**We sincerely thank the reviewers for their thorough evaluation of our paper and for considering our rebuttal when assigning their final scores.**

Best regards,
The authors

---

### Meta-Review · Area_Chair_QHn4 · 2024-12-22

**Metareview:**

The paper has proposed a novel algorithm with strong convergence guarantees. I concur with the positive evaluation of the reviewers. Recommend accept.

**Additional Comments On Reviewer Discussion:**

NA

---

### Decision · Program_Chairs · 2025-01-22

Accept (Poster)